# Mitigating Information Loss in Tree-Based Reinforcement Learning via Direct Optimization

**Sascha Marton**[1][*]  **Tim Grams**[2][*]  **Florian Vogt**[1]  **Stefan Lüdtke**[3]
**Christian Bartelt**[2]  **Heiner Stuckenschmidt**[1]

[1]University of Mannheim  [2]Technical University of Clausthal  [3]University of Rostock
sascha.marton@uni-mannheim.de  tim.nico.grams@tu-clausthal.de

## Abstract

Reinforcement learning (RL) has seen significant success across various domains, but its adoption is often limited by the black-box nature of neural network policies, making them difficult to interpret. In contrast, symbolic policies allow representing decision-making strategies in a compact and interpretable way. However, learning symbolic policies directly within on-policy methods remains challenging. In this paper, we introduce SYMPOL, a novel method for SYMbolic tree-based on-POLicy RL. SYMPOL employs a tree-based model integrated with a policy gradient method, enabling the agent to learn and adapt its actions while maintaining a high level of interpretability. We evaluate SYMPOL on a set of benchmark RL tasks, demonstrating its superiority over alternative tree-based RL approaches in terms of performance and interpretability. Unlike existing methods, it enables gradient-based, end-to-end learning of interpretable, axis-aligned decision trees within standard on-policy RL algorithms. Therefore, SYMPOL can become the foundation for a new class of interpretable RL based on decision trees. Our implementation is available under: https://github.com/s-marton/sympol

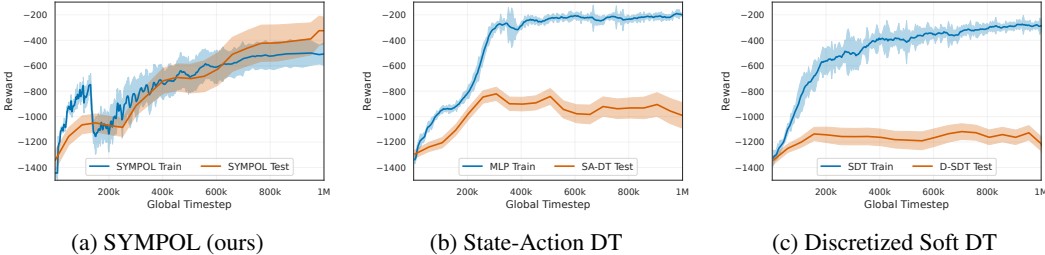

| (a) SYMPOL (ours) | (b) State-Action DT | (c) Discretized Soft DT |
|---|---|---|

**Figure 1: Information Loss in Tree-Based Reinforcement Learning on Pendulum.** Existing methods for symbolic, tree-based RL (Figure 1b and 1c) suffer from severe information loss when converting the differentiable policy used for training (e.g., the MLP for SA-DT) into the symbolic policy used for interpretation (i.e., the DT). Using SYMPOL (Figure 1a), we can directly optimize the symbolic policy with PPO and therefore have no information loss during the application.

## 1 Introduction

**Reinforcement learning lacks transparency.** Reinforcement learning (RL) has achieved remarkable success in solving complex sequential decision-making problems, ranging from robotics and autonomous systems to game playing and recommendation systems. However, the policies learned by traditional RL algorithms, represented by Neural Networks (NNs), often lack interpretability and transparency, making them difficult to understand, trust, and deploy in safety-critical or high-stakes scenarios (Landajuela et al., 2021).

---

[*]Equal Contribution

**Symbolic policies increase trust.** Symbolic policies, on the other hand, offer a promising alternative by representing decision-making strategies in terms of RL policies as compact and interpretable structures (Guo et al., 2024). These symbolic representations do not only facilitate human understanding and analysis but also ensure predictable and explainable behavior, which is crucial for building trust and enabling effective human-AI collaboration. Moreover, the deployment of symbolic policies in safety-critical systems, such as autonomous vehicles or industrial robots, could significantly improve their reliability and trustworthiness. By providing human operators with a clear understanding of the decision-making process, symbolic policies can facilitate effective monitoring, intervention, and debugging, ultimately enhancing the safety and robustness of these systems. In this context, decision trees (DTs) are particularly effective as symbolic policies for RL, as their hierarchical structure provides natural interpretability.

**Existing challenges.** Despite these promising prospects, the field of symbolic RL faces several challenges. One main reason is given by the fact that many symbolic models, like DTs, are non-differentiable and cannot be integrated in existing RL frameworks. Therefore, traditional methods for learning symbolic policies often rely on custom and complex training procedures (Costa et al., 2024; Vos & Verwer, 2023; Kanamori et al., 2022), limiting their applicability and scalability. Alternative methods involve pre-trained NN policies combined with some post-processing to obtain an interpretable model (Silva et al., 2020; Liu et al., 2019; 2023; Bastani et al., 2018). However, post-processing introduces a mismatch between the optimized policy and the model obtained for interpretation, which can lead to loss of crucial information, as we show in Figure 1.

**Contribution.** To mitigate the impact of information loss, a direct optimization of the policy is crucial. Existing methods that employ a direct optimization of a tree-based policy, such as those described by Silva et al. (2020) learn differentiable, soft decision trees which do not provide a high level of interpretability. To obtain interpretable, axis-aligned DTs, these methods require post-hoc distillation or discretization and therefore suffer from information loss (see Figure 1). In this paper, we introduce SYMPOL, SYMbolic tree-based on-POLicy RL, a novel method emplaying a direct optimization axis-aligned DT policies end-to-end. Our contributions are as follows:

- We integrate GradTree (Marton et al., 2024a) into existing RL frameworks via a separate actor-critic architecture to directly optimize DT policies and extend it to continuous action spaces (Section 4.1).

- We propose a dynamic rollout buffer to enhance exploration stability and a dynamic batch size through gradient accumulation to improve gradient stability (Section 4.2) to mitigate the instability of DT training in dynamic environments.

- We propose using weight decay on a subset of parameters to support a dynamic adjustment of the model parameters when optimizing DTs with gradient descent (Section 4.1).

As a result, SYMPOL does not depend on pre-trained NN policies, complex search procedures, or post-processing steps, but can be seamlessly integrated into existing RL algorithms (Section 3).

**Results.** Through extensive experiments on benchmark RL environments, we demonstrate that SYMPOL does not suffer from information loss and outperforms existing tree-based RL approaches in terms of interpretability and performance (Section 5.2), providing human-understandable explanations. In most environments, SYMPOL's performance is comparable to full-complexity models, while in categorical environments, it even surpasses them. Furthermore, we provide a case study (Section 6) to show how interpretable policies help in detecting misbehavior and misgeneralization which might remain unnoticed with commonly used black-box policies.

## 2 RELATED WORK

Recently, the integration of symbolic methods into RL has gained significant attention. Symbolic RL does cover different approaches including program synthesis (Trivedi et al., 2021; Penkov & Ramamoorthy, 2019; Verma et al., 2018), concept bottleneck models (Ye et al., 2024), piecewise linear networks (Wabartha & Pineau, 2024) and mathematical expressions (Landajuela et al., 2021; Guo et al., 2024; Luo et al., 2024; Kamienny et al., 2022). Another line of work aims to synthesize symbolic policies using logical rules, leveraging for instance differentiable inductive logic programming for gradient-based optimization (Delfosse et al., 2024b; Jiang & Luo, 2019; Cao et al., 2022).

In contrast to first-order rules, DTs offer greater flexibility by not only combining atomic conditions but also comparing features against thresholds — a critical capability for handling continuous observation spaces. Furthermore, trees have also been in used in other agentic components than the policy, such as reward functions (Milani et al., 2022; Kalra & Brown, 2023; 2022). In this paper, we focus exclusively on tree-based methods for symbolic RL. Similarly, ensemble methods (Fuhrer et al., 2024; Min & Elliott, 2022) have been proposed. However, policies consisting of hundreds of trees and nodes lack interpretability and therefore are out of scope for this paper. Several approaches have been proposed to leverage the strengths of interpretable, tree-based representations within RL frameworks, each approach comes with its own critical limitations. We summarize existing methods into three streams of work:

**(1) Post-processing.** One line learns full-complexity policies first and then performs some kind of post-processing for interpretability. One prominent example is the VIPER algorithm (Bastani et al., 2018) where a NN policy is learned before distilling a DT from the policy. However, distillation methods often suffer from significant performance mismatches between the training and evaluation policies (Figure 1b). To mitigate this mismatch, existing methods often learn large DTs (VIPER learns DTs with 1,000 nodes) and therefore aim for systematic verification rather than interpretability. In contrast, SYMPOL is able to learn small, interpretable DTs (average of only 50 nodes) without information loss. Following VIPER, various authors proposed similar distillation methods (Li et al., 2021; Liu et al., 2019; 2023; Jhunjhunwala et al., 2020). Furthermore, Kohler et al. (2024) propose a novel method that distills interpretable and editable programmatic tree policies. In contrast to SYMPOL, the extracted trees are not considered axis-aligned, as they allow for linear combinations and multiple features within the internal nodes.

**(2) Custom optimization.** Methods involving custom, tree-specific optimization techniques and/or objectives (Ernst et al., 2005; Roth et al., 2019; Gupta et al., 2015; Kanamori et al., 2022) are generally more time-consuming and less flexible. As a result, their policy models cannot be easily integrated into existing learning RL frameworks. Examples are evolutionary methods (Costa et al., 2024; Custode & Iacca, 2023) and linear integer programming (Vos & Verwer, 2023). Topin et al. (2021) propose Iterative Bounding Markov Decision Process (IBMDP) that allow learning DT policies through a masking procedure and modified value updates by using arbitrary function approximators. However, using IBMDP, the learning problem becomes more complex compared to the base MDP, which can result in poor scalability and limits the applicability to very simple tasks (Milani et al., 2022; Kohler et al., 2024). In contrast, SYMPOL optimizes a DT policy directly on the base MDP, avoiding these limitations.

**(3) Soft Decision Trees (SDTs).** Methods optimizing SDTs (Silva et al., 2020; Silva & Gombolay, 2021; Coppens et al., 2019; Tambwekar et al., 2023; Liu et al., 2022; Farquhar et al., 2017) are difficult to interpret since they usually involve multiple features simultaneously at each decision node, creating complex, multidimensional splits rather than straightforward, single-feature thresholds. Nevertheless, the trees are usually not easily interpretable and techniques such as discretizing the learned trees into more interpretable representations are applied (Silva et al., 2020), occasionally resulting in high performance mismatches (Figure 1c). In contrast, SYMPOL directly optimizes hard, axis-aligned DTs and therefore does not exhibit a performance loss (Figure 1a).

**Distinction of SYMPOL from Differentiable and Soft Decision Trees.** In existing work like Silva et al. (2020), differentiable decision trees typically correspond to SDTs, achieving differentiability by relaxing discrete decisions in terms of feature selection at each internal node and path selection. This approach is fundamentally different from SYMPOL, which does *not* use differentiable decision trees. Instead, SYMPOL leverages GradTree to optimize standard, non-differentiable decision trees through gradient descent, as we will show in Section 4.

## 3 PRELIMINARIES

**Markov Decision Process (MDP).** We study a deterministic MDP $(\mathcal{S}, \mathcal{A}, \mathcal{P}, r, \gamma)$ where $\mathcal{S}$ and $\mathcal{A}$ are finite state and action spaces, $\mathcal{P} \colon \mathcal{S} \times \mathcal{A} \times \mathcal{S} \to [0, 1]$ defines the transition dynamics, $r \colon \mathcal{S} \times \mathcal{A} \to \mathbb{R}$ is the reward function and $\gamma$ the discount factor. At each timestep $t$, an agent samples an action $a_t$ from policy $\pi! \colon \mathcal{S} \to \mathcal{A}$ based on observation $s_t \in \mathcal{S}$ and executes it, receiving reward $r_t$. The value function $\mathcal{V}^\pi(s) = \mathbb{E}_{a_t \sim \pi, s_{t+1} \sim \mathcal{P}} \left[ \sum_{t=0}^{\infty} \gamma^t r(s_t, a_t) \mid s_t = s \right]$ approximates the expected return when starting in state $s$ and then acting according to policy $\pi$. Similarly, the action-value function

$\mathcal{Q}^\pi(s, a) = \mathbb{E}_{a_t \sim \pi, s_{t+1} \sim \mathcal{P}} \left[ \sum_{t=0}^{\infty} \gamma^t r(s_t, a_t) \mid s_t = s, a_t = a \right]$ estimates the expected return when selecting action $a$ in state $s$ and then following policy $\pi$. Finally, the advantage function $\mathcal{A}^\pi(s, a) = \mathcal{Q}^\pi(s, a) - \mathcal{V}^\pi(s)$ defines the difference between the expected return when choosing action $a$ in state $s$ and the expected return when following the policy $\pi$ from state $s$. Overall, we aim for finding an optimal policy $\pi^*$ that maximizes the expected discounted return $J(\pi) = \mathbb{E} \left[ \sum_{t=0}^{\infty} \gamma^t r(s_t, a_t) \right]$.

**Proximal Policy Optimization (PPO).** PPO (Schulman et al., 2017) is an on-policy, actor-critic RL method designed to enhance the training stability. The algorithm introduces a clipped surrogate objective to restrict the policy update step size. The main idea is to constrain policy changes to a small trust region, preventing large updates that could destabilize training. Formally, PPO optimizes:

$$\mathcal{L}^{\text{CLIP}}(\theta) = \mathbb{E}_{a_t \sim \pi_{\theta_{\text{old}}}, s_{t+1} \sim \mathcal{P}} \left[ \min \left( \frac{\pi_\theta(a_t|s_t)}{\pi_{\theta_{\text{old}}}(a_t|s_t)} \hat{A}_t, \quad \text{clip} \left( \frac{\pi_\theta(a_t|s_t)}{\pi_{\theta_{\text{old}}}(a_t|s_t)}, 1 - \epsilon, 1 + \epsilon \right) \hat{A}_t \right) \right] \quad (1)$$

where $\frac{\pi_\theta(a_t|s_t)}{\pi_{\theta_{\text{old}}}(a_t|s_t)}$ is the probability ratio between the new policy $\pi_\theta$ and old policy $\pi_{\theta_{\text{old}}}$. $\hat{A}_t$ is an estimate of the advantage function at time step $t$ and $\epsilon$ is a hyperparameter for the clipping range.

## 4 SYMPOL: SYMBOLIC ON-POLICY RL

In the following, we formalize the online training of hard, axis-aligned DTs with the PPO objective. Therefore, SYMPOL utilizes GradTree (Marton et al., 2024a) as a core component to learn a DT policy directly from policy gradients. In contrast to existing work on RL with DTs, this allows an optimization of the DT on-policy without information loss. The main conceptual difference to existing work that learn symbolic policies end-to-end (Delfosse et al., 2024b; Fuhrer et al., 2024; Delfosse et al., 2024c; Topin et al., 2021; Luo et al.) is that SYMPOL does *not* require any modification of the RL framework itself, making the proposed method framework-agnostic. As a result, interpretable policies with SYMPOL are learned in the same way as NN policies are commonly learned. In the main paper, we focus on PPO as the, we believe, most prominent on-policy RL method. To support our claim of seamless integration, we provide additional results using Advantage Actor-Critic (A2C) (Mnih et al., 2016) in Appendix A.1 To efficiently learn DT policies with SYMPOL, we employed several crucial (see ablation study in Table 5) modifications, which we will elaborate below.

### 4.1 LEARNING DTS WITH POLICY GRADIENTS

**Arithmetic DT policy formulation.** Traditionally, DTs involve nested concatenations of rules. In GradTree, DTs are formulated as arithmetic functions based on addition and multiplication to facilitate gradient-based learning. Therefore, our resulting DT policy is fully-grown (i.e., complete, full) and can be pruned post-hoc. Our basic pruning involves removing redundant paths, which significantly reduces the complexity. We define a path as redundant if the decision is already determined either by previous splits or based on the range of the selected feature. More details are given in Appendix A.4. Overall, we formulate a DT policy $\pi$ of depth $d$ with respect to its parameters as:

$$\pi(\boldsymbol{s}|\boldsymbol{a}, \boldsymbol{\tau}, \boldsymbol{\iota}) = \sum_{l=0}^{2^d - 1} a_l \, \mathbb{L}(\boldsymbol{s}|l, \boldsymbol{\tau}, \boldsymbol{\iota}) \quad (2)$$

where $\mathbb{L}$ is a function that indicates whether a state $\boldsymbol{s} \in \mathbb{R}^{|\mathcal{S}|}$ belongs to a leaf $l$, $\boldsymbol{a} \in \mathcal{A}^{2^d}$ denotes the selected action for each leaf node, $\boldsymbol{\tau} \in \mathbb{R}^{2^d - 1}$ represents split thresholds and $\boldsymbol{\iota} \in \mathbb{N}^{2^d - 1}$ the feature index for each internal node.

**Dense architecture.** To support a gradient-based optimization and ensure an efficient computation via matrix operations, we make use of a dense DT representation. Traditionally, the feature index vector $\boldsymbol{\iota}$ is one-dimensional. However, as in GradTree, we expand it into a matrix form. Specifically, this representation one-hot encodes the feature index, converting $\boldsymbol{\iota} \in \mathbb{R}^{2^d - 1}$ into a matrix $\boldsymbol{I} \in \mathbb{R}^{(2^d - 1) \times |\mathcal{S}|}$. Similarly, for split thresholds, instead of a single value for all features, individual values for each feature are stored, leading to $\boldsymbol{T} \in \mathbb{R}^{(2^d - 1) \times |\mathcal{S}|}$. The dense representation is visualized in Figure 2. Please note, that in contrast to SDTs, the dense representation of SYMPOL corresponds to an equivalent standard DT representation at each point in time, ensuring that the underlying model is a hard, axis-aligned DT. By enumerating the internal nodes in breadth-first order,

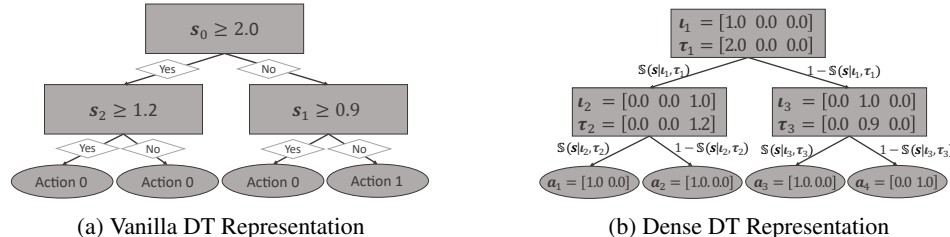

(a) Vanilla DT Representation          (b) Dense DT Representation

Figure 2: **Standard vs. Dense DT Representation.** A comparison between the standard decision tree representation and its dense equivalent, illustrated using an example decision tree of depth 2, with a state space of dimensionality 3 and two possible actions.

we can redefine the indicator function $\mathbb{L}$ for a leaf $l$, resulting in

$$\pi(\boldsymbol{s}|\boldsymbol{a}, T, I) = \sum_{l=0}^{2^d-1} a_l \, \mathbb{L}(\boldsymbol{s}|l, \boldsymbol{T}, \boldsymbol{I}) \tag{3}$$

$$\text{where} \quad \mathbb{L}(\boldsymbol{s}|l, \boldsymbol{T}, \boldsymbol{I}) = \prod_{j=1}^{d} \left(1 - \mathfrak{p}(l,j)\right) \mathbb{S}(\boldsymbol{s}|\boldsymbol{I}_{\mathfrak{i}(l,j)}, \boldsymbol{T}_{\mathfrak{i}(l,j)}) + \mathfrak{p}(l,j) \left(1 - \mathbb{S}(\boldsymbol{s}|\boldsymbol{I}_{\mathfrak{i}(l,j)}, \boldsymbol{T}_{\mathfrak{i}(l,j)})\right) \tag{4}$$

Here, $\mathfrak{i}$ is the index of the internal node preceding a leaf node $l$ at a certain depth $j$ and $\mathfrak{p}$ indicates whether the left ($\mathfrak{p} = 0$) or the right branch ($\mathfrak{p} = 1$) was taken.

**Axis-aligned splitting.** Typically, DTs use the Heaviside function for splitting, which is non-differentiable. We use the split function introduced in GradTree to account for reasonable gradients:

$$\mathbb{S}(\boldsymbol{s}|\boldsymbol{\iota}, \boldsymbol{\tau}) = \lfloor S\left(\boldsymbol{\iota} \cdot \boldsymbol{s} - \boldsymbol{\iota} \cdot \boldsymbol{\tau}\right) \rceil \tag{5}$$

where $S(z) = \frac{1}{1+e^{-z}}$ represents the logistic function, $\lfloor \cdot \rceil$ stands for rounding and $\boldsymbol{a} \cdot \boldsymbol{b}$ denotes the dot product. We further need to ensure that $\boldsymbol{\iota}$ is a one-hot encoded vector to account for axis-aligned splits. This is achieved by applying a hardmax transformation before calculating $\mathbb{S}$. Both rounding and hardmax operations are non-differentiable and therefore, SYMPOL is *not* considered as a soft or differentiable DT method. Instead, to overcome non-differentiability, SYMPOL employs a straight-through operator (Bengio et al., 2013) during backpropagation. This allows the model to use non-differentiable operations in the forward pass while ensuring gradient propagation in the backward pass. As a result, we can directly learn an interpretable DT from policy gradient. This makes SYMPOL framework-agnostic and facilitates a seamless integration into existing RL frameworks.

**Weight decay.** In contrast to GradTree, which employs an Adam (Kingma & Ba, 2014) optimizer with stochastic weight averaging (Izmailov et al., 2018), we opted for an Adam optimizer with weight decay (Loshchilov & Hutter, 2017). In the context of SYMPOL, weight decay does not serve as a regularizer for model complexity, as the interpretation of model parameters differs. We distinguish between three types of parameters: the distributions in the leaves ($\boldsymbol{a}$), the split index encoding ($\boldsymbol{I}$), and the split values ($\boldsymbol{T}$). We do not apply weight decay to the split values because they are independent of magnitude. However, for the split indices and leaves, weight decay enhances exploration during training by penalizing large parameter values. As a result, the distribution of the split index selection and class prediction are narrow and have lower variance. This aids in dynamically adjusting which feature is considered at a split and in altering the predicted leave distribution.

**Actor-critic network architecture.** Commonly, the actor and critic use a similar network architecture or even share the same weights (Schulman et al., 2017). While SYMPOL aims for a simple and interpretable policy, we do not have the same requirements for the critic. Therefore, we decided to only employ a tree-based actor and use a full-complexity NN as a value function. As a result, we can still capture complexity through the value function, without loosing interpretability, as we maintain a simple and interpretable policy.

**Continuous action spaces.** Furthermore, we extend the DT policy of SYMPOL to environments with continuous action spaces. Therefore, instead of predicting a categorical distribution over the

classes, we predict the mean of a normal distribution at each leaf and utilize an additional variable $\sigma_{\log} \in \mathbb{R}^{|\mathcal{A}|}$ to learn the log of the standard deviation.

## 4.2 ADDRESSING TRAINING STABILITY

One main challenge when using DTs as a policy is the stability. While a stable training is also desired and often hard to achieve for a NN policy, this is even more pronounced for SYMPOL. This is mainly caused by the inherent tree-based architecture. Changing a split at the top of the tree can have a severe impact on the whole model, as it can completely change the paths taken for certain observations. This is especially relevant in the context of RL, where the data distribution can vary highly between iterations. To mitigate the impact of highly non-stationary training samples, especially at early stages of training, we made two crucial modifications for improved stability.

**Exploration stability.** Motivated by the idea that rollouts of more accurate policies contain increasingly diverse, higher quality samples, we implemented a dynamic number of environment steps between training iterations. Let us consider a pendulum as an example. While at early stages of training a relatively small sample size facilitates faster learning as the pendulum constantly flips, more optimal policies lead to longer rollouts and therefore more expressive and diverse experiences in the rollout buffer. Similarly, the increasing step counts stabilize the optimization of policy and critic, as the number of experiences for gradient computation grow with agent expertise and capture the diversity within trajectories better. Therefore, our novel collection approach starts with $n_{init}$ environment steps and expands until $n_{final}$ actions are taken before each training iteration. For computational efficiency reasons, instead of increasing the size of the rollout buffer at every time step, we introduce a step-wise exponential function. The exponential increase supports exploration in the initial iterations, while maintaining stability at later iterations. Hence, we define the number of steps in the environment $n_t$ at time step $t$ as

$$n_t = n_{\text{init}} \times 2^{\left\lfloor \frac{(t+1) \times i}{1 + t_{\text{total}}} \right\rfloor - 1} \text{ with } i = 1 + \log_2\left(\frac{n_{\text{init}}}{n_{\text{final}}}\right) \tag{6}$$

For our experiments, we define $n_{\text{init}}$ as a hyperparameter (similar to the static step size for other methods) and set $n_{\text{final}} = 128 \times n_{\text{init}}$ and therefore $i = 8$ which we observed is a good default value.

**Gradient stability.** We also utilize large batch sizes for SYMPOL resulting in less noisy gradients, leading to a smoother convergence and better stability. In this context, we implement gradient accumulation to virtually increase the batch size further while maintaining memory-efficiency. As reduced noise in the gradients also leads to less exploration in the parameter space, we implement a dynamic batch size, increasing in the same rate as the environment steps between training iterations (Equation 6). Therefore, we can benefit from exploration and fast convergence early on and increase gradient stability during the training.

## 5 EVALUATION

We designed our experiments to evaluate whether SYMPOL can learn accurate DT policies without information loss and observe whether the trees learned by SYMPOL are small and interpretable. As mentioned above, we focus on PPO as the most prominent actor-critic, on-policy RL algorithm in our evaluation. To support our claim that SYMPOL can be seamlessly integrated into existing on-policy RL frameworks, we additionally provide results using A2C in Appendix A.1.

### 5.1 EXPERIMENTAL SETTINGS

**Setup.** We implemented SYMPOL in a highly efficient single-file JAX implementation that allows a flawless integration with highly optimized training frameworks (Lange, 2022; Weng et al., 2022; Bonnet et al., 2024).[1] We evaluated our method on several environments commonly used for benchmarking RL methods. Specifically, we used the control environments CartPole (CP), Acrobot (AB), LunarLander (LL), MountainCarContinuous (MC-C) and Pendulum (PD-C), as well as the MiniGrid (Chevalier-Boisvert et al., 2023) environments Empty-Random (E-R), DoorKey (DK), LavaGap (LG) and DistShift (DS).

---

[1]Our implementation is available under `https://github.com/s-marton/sympol`.

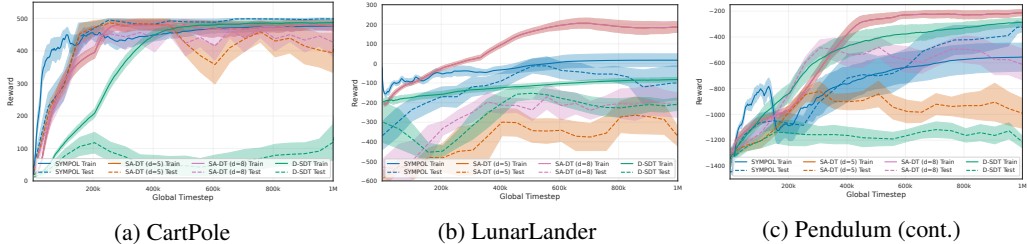

|  (a) CartPole | (b) LunarLander | (c) Pendulum (cont.) |

Figure 3: **Selected Training Curves.** Shows the training reward of the full-complexity policy (e.g. MLP in the case of SA-DT) as solid line and the test reward of the interpretable policy as dashed line for three control environments. Additional, more detailed results are in Appendix A.7.

**Methods.** The goal of this evaluation is to compare SYMPOL to alternative methods that allow an interpretation of RL policies as a symbolic, axis-aligned DTs. Building on previous work (Silva et al., 2020), we employ an MLP and an SDT to provide a reference to state-of-the-art results and evaluate two methods grounded in the interpretable RL literature:

- **State-Action DTs (SA-DT)**: SA-DTs are the most common method to generate interpretable policies post-hoc. Hereby, we first train an MLP policy, which is then distilled into a DT as a post-processing step after the training. SA-DT can be considered as a version of DAGGER (Ross et al., 2011) and therefore a simplified version of VIPER (Bastani et al., 2018). In a comparative experiment (see Appendix A.3), we showed that for the case of learning small, interpretable DTs the performance of SA-DT is similar to those of VIPER, which is in-line with results reported e.g. by Kohler et al. (2024).

- **Discretized Soft DTs (D-SDT)**: SDTs allow gradient computation by assigning probabilities to each node. While SDTs exhibit a hierarchical structure, they are usually considered as less interpretable, since multiple features are considered in a single split and the whole tree is traversed simultaneously (Marton et al., 2024a). Therefore, Silva et al. (2020) use SDTs as policies which are discretized post-hoc to allow an easy interpretation.

**Evaluation procedure.** We report the average undiscounted cumulative reward over 5 random trainings with 5 random evaluation episodes each (=25 evaluations for each method). We trained each method for 1mio timesteps. For SYMPOL, SDT and MLP, we optimized the hyperparameters based on the validation reward with optuna (Akiba et al., 2019) for 60 trials using a predefined grid. For D-SDT we discretized the SDT and for SA-DT, we distilled the MLP with the highest performance. More details on the hyperparameters can be found in Appendix C.

## 5.2 RESULTS

**SYMPOL does not exhibit information loss.** Existing methods for learning DT policies usually involve post-processing to obtain the interpretable model. Therefore, they introduce a mismatch between the optimized and interpreted policy, which can result in information loss. The main advantage of SYMPOL is the direct optimization of a DT policy, which guarantees that there is no information loss between the optimized and interpreted policy. To show this, we calculated Cohen's D to measure the effect size comparing the validation reward of the trained model with the test reward of the applied, optionally post-processed model (Table 1). We can observe very large effects for SA-DT and D-SDT and only a very small effect for SYMPOL, similar to full-complexity models MLP and SDT. This discrepancy can also be observed in the training curves in Figure 3.

Table 1: **Information Loss**. We calculated Cohen's D to measure effect size between the validation reward of the trained and the test reward of the applied model. Values $> 0.8$ are considered as a large effect. Detailed results are in Appendix A.2

|  | Cohen's D $\downarrow$ |
| --- | --- |
| SYMPOL (ours) | **-0.019** |
| SA-DT (d=5) | 3.449 |
| SA-DT (d=8) | 2.527 |
| D-SDT | 3.126 |
| MLP | 0.306 |
| SDT | 0.040 |

**SYMPOL learns accurate DT policies.** We evaluated our approach against existing methods on control environments in Table 2. SYMPOL is consistently among the best interpretable models and achieves significantly higher rewards compared to alternative methods for learning DT policies on several environments, especially on LL and PD-C. Further, SYMPOL consistently solves CP and AB and is competitive to full-complexity models on most environments.

Table 2: **Control Performance.** We report the average undiscounted cumulative test reward over 25 random trials. The best interpretable method, and methods not statistically different, are marked bold.

|  | CP | AB | LL | MC-C | PD-C |
|---|---|---|---|---|---|
| SYMPOL (ours) | **500** | **- 80** | **- 57** | **94** | **- 323** |
| D-SDT | 128 | -205 | -221 | -10 | -1343 |
| SA-DT (d=5) | 446 | -97 | -197 | **97** | -1251 |
| SA-DT (d=8) | 476 | **- 75** | -150 | **96** | - 854 |
| MLP | 500 | - 72 | 241 | 95 | - 191 |
| SDT | 500 | - 77 | -124 | - 4 | - 310 |

**DT policies offer a good inductive bias for categorical environments.** While SYMPOL achieves great results in control benchmarks, it may not be an ideal method for environments modeling physical relationships. As recently also noted by Fuhrer et al. (2024), tree-based models are best suited for categorical environments due to their effective use of axis-aligned splits. In our experiments on MiniGrid (Table 3), SYMPOL achieves comparable or superior results to full-complexity models (e.g. on LG-7). The performance gap be-

Table 3: **MiniGrid Performance.** We report the average undiscounted cumulative test reward over 25 random trials. The best interpretable method, and methods not statistically different, are marked bold.

|  | E-R | DK | LG-5 | LG-7 | DS |
|---|---|---|---|---|---|
| SYMPOL (ours) | **0.964** | **0.959** | **0.951** | **0.953** | 0.939 |
| D-SDT | 0.662 | 0.654 | 0.262 | 0.381 | 0.932 |
| SA-DT (d=5) | 0.583 | **0.958** | **0.951** | 0.458 | **0.952** |
| SA-DT (d=8) | 0.845 | **0.961** | **0.951** | 0.799 | **0.954** |
| MLP | 0.963 | 0.963 | 0.951 | 0.760 | 0.951 |
| SDT | 0.966 | 0.959 | 0.839 | 0.953 | 0.954 |

tween SA-DT and SYMPOL is smaller in certain MiniGrid environments due to less complex environment transition functions and missing randomness, making the distillation easy. Considering more complex environments with randomness or lava like E-R or LG-7, SYMPOL outperforms alternative methods by a substantial margin.

**DT policies learned with SYMPOL are small and interpretable.** While we trained SYMPOL with a depth of 7 and therefore 255 possible nodes, the effective tree size after pruning is significantly smaller with only 50.5 nodes (internal and leaf combined) on average. This can be attributed to a self-pruning mechanism that is inherently applied by SYMPOL in learning redundant paths during the training and therefore only optimizing relevant parts. Furthermore, DTs learned with SYMPOL are smaller than SA-DTs (d=5) with an average of 60.3 nodes and significantly

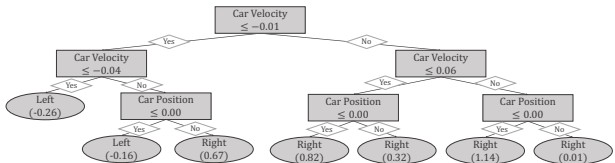

Figure 4: **SYMPOL Policy for MC-C.** The main rule encoded by this DT is that the car should accelerate to the left, if its velocity is negative, and to the right if it is positive. This essentially increases the speed of the car over time, making it possible to reach the goal at the top of the hill. The magnitude of acceleration is mainly determined by the current position, reducing the action cost.

smaller than SA-DTs (d=8) averaging 291.6 nodes. The pruned D-SDTs are significantly smaller with only 16.5, but also have a very poor performance, as shown in the previous experiment. An exemplary DT learned by SYMPOL is visualized in Figure 4. Extended results, including a comparison with SDTs are in Appendix A.4 and A.5.

**SYMPOL is efficient.** In RL, the actor-environment interaction frequently constitutes a significant portion of the total runtime. For smaller policies, in particular, the runtime is mainly determined by the time required to execute actions within the environment to obtain the next observation, while the time required to execute the policy itself having a comparatively minimal impact on runtime. Therefore, recent research put much effort into optimizing this interaction through environment

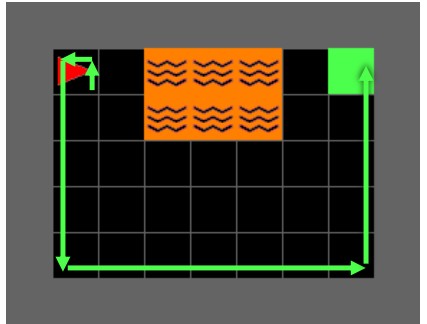

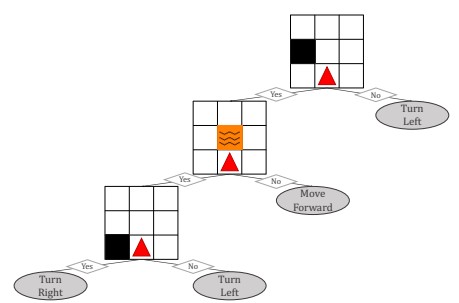

Figure 6: **DistShift.** We show the training environment for the agent along with the starting position and goal. The path taken by SYMPOL (see Figure 7) is marked by green arrows and solves the environment.

Figure 7: **SYMPOL Policy.** This image shows the DT policy of SYMPOL. Split nodes are visualized as the 3x3 view grid of the agent with one square marking the considered object and position. If the visualized object is present at this position, the true path (left) is taken.

vectorization. The design of SYMPOL, in contrast to existing methods for tree-based RL, allows a seamless integration with these highly efficient training frameworks. As a result, the runtime of SYMPOL is almost identical to using an MLP or SDT as policy, averaging less than 30 seconds for 1mio timesteps. Detailed results are in Appendix A.7.

**Ablation study.** In Section 4, we introduced several crucial components to facilitate an efficient and stable training of SYMPOL. To support the intuitive justifications for our modifications, we performed an ablation study (Figure 5) to evaluate the relevance of the individual components. Our results confirm that each component substantially contributes to the overall performance.

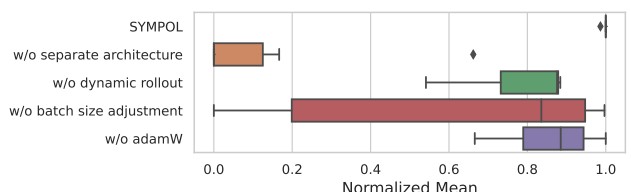

Figure 5: **Ablation Study**. We report the mean normalized reward over all control environments (details in Table 11).

## 6 CASE STUDY: DETECTING GOAL MISGENERALIZATION

To demonstrate the benefits of SYMPOLs enhanced transparency, we present a case study on goal misgeneralization (Di Langosco et al., 2022). Good policy generalization is vital in RL, yet agents often exhibit poor out-of-distribution performance, even with minor environmental changes. Goal misgeneralization is a well-researched out-of-distribution robustness failure that occurs when an agent learns robust skills during training but follows unintended goals. This happens when the agent's behavioral objective diverges from the intended objective, leading to high rewards during training but poor generalization during testing. For instance, NNs were shown to systematically misgeneralize on Atari environments (Farebrother et al., 2018; Delfosse et al., 2024a).

To demonstrate that SYMPOL can help in detecting misaligned behavior, let us consider the Dist-Shift environment from MiniGrid, shown in Figure 6. The environment is designed to test for misgeneralization (Chevalier-Boisvert et al., 2023), as the goal is repeatedly placed in the top right corner and the lava remains at the same position. We can formulate the intended behavior according to the task description as avoiding the lava and reaching a specific goal location. SYMPOL, similar to other methods, solved the task consistently. The advantage of SYMPOL is the tree-based structure, which is easily interpretable. When inspecting the SYMPOL policy (Figure 7), we can immediately observe that the agent has not captured the actual task correctly. Essentially, it has only learned to keep an empty space on the left of the agent (which translates into following the wall) and not to step into lava (but not to get around it). While this is sufficient to solve this exact environment, it is evident, that the agent has not generalized to the overall goal.

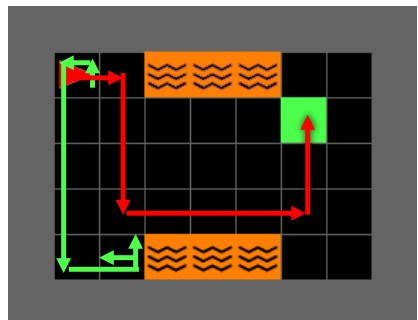

Figure 8: **DistShift with Domain Randomization.** This is a modified version of DistShift, with goal and lava at random positions. SYMPOL (Figure 7), visualized as the green line, is not able to solve the randomized environment. Training SYMPOL with domain randomization (Figure 9), visualized as the red line, is able to solve the environment.

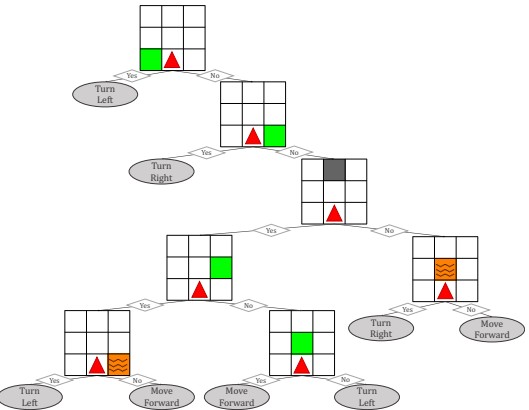

Figure 9: **SYMPOL Policy with Domain Randomization.** The SYMPOL policy (Figure 9) retrained with domain randomization. The agent now has learned to avoid lava and walls, as well as identifying and walk into the goal.

In order to test for misgeneralization, we created test environments in which the agent has to reach a random goal placed with lava placed at a varying locations. As already identified based on the interpretable policy, we can observe in Figure 8 that the agent gets stuck when the goal or lava positions change. Alternative non-interpretable policies exhibit the same behavior, which might remain unnoticed due to the black-box nature. Instead of simply looking at the learned policy with SYMPOL, alternative methods would require using external methods or designing complex test cases to detect such misbehavior. Alternative methods to generate DT policies like SA-DT also provide an interpretable policy, but as already shown during our experiments, frequently come with severe information loss. Due to this information loss, we cannot ensure that we are actually interpreting the policy, which is guaranteed using SYMPOL.

Based on these insights, we retrained SYMPOL with domain randomization. The resulting policy (see Figure 9) now solves the randomized environments (see Figure 8), still maintaining interpretability. In line with our results, Delfosse et al. (2024c) showed that using interpretable DT policies can help to mitigate goal misgeneralization which showcases the potential benefit of using interpretable RL policies to ensure a good generalization.

## 7 CONCLUSION AND FUTURE WORK

In this paper, we introduced SYMPOL, a novel method for tree-based RL. SYMPOL can be seamlessly integrated into existing on-policy RL frameworks, where the DT policy is directly optimized on-policy while maintaining interpretability. This direct optimization guarantees that the explanation exactly matches the policy learned during training, avoiding the information loss often encountered with existing methods that rely on post-processing to obtain an interpretable policy. Furthermore, the performance of interpretable DT policies learned by SYMPOL is significantly higher compared to existing methods, particularly in environments involving more complex environment transition functions or randomness. We believe that SYMPOL represents a significant step towards bridging the gap between the performance of on-policy RL and the interpretability and transparency of symbolic approaches, paving the way for the widespread adoption of trustworthy and explainable AI systems in safety-critical and high-stakes domains.

While we focused on an actor-critic on-policy RL method, the flexibility of SYMPOL allows an integration into arbitrary policy optimization RL frameworks including off-policy methods like Soft Actor-Critic (SAC) (Haarnoja et al., 2018) or Advantage Weighted Regression (AWR) (Peng et al., 2019), which can be explored in future work. Also, it would be interesting to evaluate a more complex, forest-like tree structure as a performance-interpretability trade-off, similar to Marton et al. (2024b), especially based on the promising results of Fuhrer et al. (2024) for tree-based RL.

ACKNOWLEDGMENTS

This research was supported in part by the German Federal Ministry for Economic Affairs and Climate Action of Germany (BMWK), and in part by the German Federal Ministry of Education and Research (BMBF).

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

# A    ADDITIONAL RESULTS

In this section, we present additional results to support the claims made in the main paper, along with extended results for the summarizing tables. We focus on the control environments because they offer a diverse suite of benchmarks that cover different tasks and include both continuous and discrete action spaces. We chose not to include the MiniGrid environments here because their inclusion could distort the results, particularly the averages calculated in the main paper, as all MiniGrid environments involve similar tasks and feature a discrete action and observation space. The primary reason for including MiniGrid in the main paper is to provide additional experimental results that confirm the robustness and applicability of our method across different domains, as well as to highlight that tree-based methods offer a beneficial inductive bias for these categorical environments.

## A.1    EVALUATION WITH ALTERNATIVE RL ALGORITHMS: ADVANTAGE ACTOR CRITIC (A2C)

To support our claim that SYMPOL can be integrated into arbitrary on-policy frameworks, we provide results on A2C in the following. The reported results use optimized hyperparameters for each method. In general, A2C is considered as less stable compared to PPO, which is an additional challenge for SYMPOL, as training stability is especially crucial for DT policies. As A2C does not update the policy in minibatches over multiple epochs, we did not include a dynamic batch size here, but update SYMPOL with a single update over the rollout to stay consistent with the A2C algorithm.

Table 4: **A2C Control Performance.** We report the average undiscounted cumulative test reward over 25 random trials. The best interpretable method, and methods not statistically different, are marked bold.

|  | CP | AB | LL | MC-C | PD-C |
|---|---|---|---|---|---|
| SYMPOL (ours) | **500** | **- 84** | **- 85** | **58** | **- 502** |
| D-SDT | 11 | -427 | -396 | -0 | -1137 |
| SA-DT (d=5) | 295 | -102 | -348 | 0 | -1467 |
| SA-DT (d=8) | 223 | - 99 | -367 | 2 | -1526 |
| MLP | 500 | - 78 | 208 | 0 | - 202 |
| SDT | 500 | - 85 | -159 | 0 | - 201 |

Table 5: **A2C Control Performance Comparison.** We report the average undiscounted cumulative test reward over 25 random trials, comparing A2C with PPO using optimized hyperparameters. A number is marked bold if the performance achieved with the underlying RL algorithm (PPO or A2C) is significantly better or not statistically different from the best result.

|  | MLP | | SDT | | SYMPOL (ours) | | SA-DT (d=5) | | SA-DT (d=8) | | D-SDT | |
|---|---|---|---|---|---|---|---|---|---|---|---|---|
|  | A2C | PPO | A2C | PPO | A2C | PPO | A2C | PPO | A2C | PPO | A2C | PPO |
| CP | **500** | **500** | **500** | **500** | **500** | **500** | 295 | **446** | 223 | **476** | 11 | **128** |
| AB | -78 | **-72** | -85 | **-77** | -84 | **-80** | -102 | **-97** | -99 | **-75** | -427 | **-205** |
| LL | 208 | 241 | -159 | -124 | -85 | **-57** | -348 | **-197** | -367 | **-150** | -396 | **-221** |
| MC-C | 0 | **95** | **0** | -4 | 58 | **94** | 0 | **97** | -2 | **96** | **0** | -10 |
| PD-C | -202 | **-191** | **-201** | -310 | -502 | **-323** | -1467 | **-1251** | -1526 | **-854** | **-1137** | -1343 |

We compared the performance of all methods using A2C on control environments in Table 4, and additionally provided a direct comparison between PPO and A2C in Table 5. When using A2C, SYMPOL consistently outperforms other interpretable models. The performance gap becomes even more pronounced when using A2C instead of PPO, as SYMPOL achieves substantially higher performance than all other interpretable models in each environment. On MC-C, SYMPOL is the only method that achieves a positive reward, whereas even full-complexity models were unable to solve the task. This can be attributed to the lower training stability of A2C compared to PPO. This could also explain the poor results of distillation methods, as the policy learned by full-complexity models, even when achieving a high test reward, is potentially less consolidated, making it harder to distill.

However, to confirm this assumption, further experiments would be required. Based on these results, we can confirm that SYMPOL can seamless be integrated into other RL algorithms, demonstrating the high flexibility of our proposed method. Additionally, our method can benefit from advances in RL, as it can be seamlessly integrated into novel frameworks.

## A.2 INFORMATION LOSS

We provide detailed results on the information loss which can result as a consequence of discretization (for D-SDT) or distillation (for SA-DT). In Table 6, we report the validation reward of the trained model along with the test reward of the discretized model. We can clearly observe that there are major differences for SA-DT and D-SDT on several datasets, indicating information loss. In Table 7, we report Cohen's D to measure the effect size comparing the validation reward of the trained model with the test reward of the applied, optionally post-processed model. Again, we can clearly see large effects for SA-DT and D-SDT on several datasets, especially for PD-C and LL, but also CP. Furthermore, the training curves in Figure 14 visually show the information loss during the training.

Table 6: **Information Loss (Comparison)**. We report the validation reward of the trained model and the test reward of the applied model.

|  | MLP | | SDT | | SYMPOL (ours) | | SA-DT (d=5) | | SA-DT (d=8) | | D-SDT | |
|---|---|---|---|---|---|---|---|---|---|---|---|---|
|  | valid | test | valid | test | valid | test | valid | test | valid | test | valid | test |
| CP | 500 | 500 | 500 | 500 | 500 | 500 | 500 | 446 | 500 | 476 | 500 | 128 |
| AB | -71 | -72 | -89 | -77 | -79 | -80 | -71 | -97 | -71 | -75 | -89 | -205 |
| LL | 256 | 241 | -91 | -124 | -9 | -57 | 256 | -197 | 256 | -150 | -91 | -221 |
| MC-C | 95 | 95 | -4 | -4 | 87 | 94 | 95 | 97 | 95 | 96 | -4 | -10 |
| PD-C | -169 | -191 | -295 | -310 | -305 | -323 | -169 | -1251 | -169 | -854 | -295 | -1343 |

Table 7: **Information Loss (Cohen's D)**. We calculated Cohen's D to measure effect size between the validation reward of the trained model and the test reward of the applied model. Typically, values $> 0.5$ are considered a medium and values $> 0.8$ a large effect. positive effects that are at least medium are marked as bold.

|  | MLP | SDT | SYMPOL (ours) | SA-DT (d=5) | SA-DT (d=8) | D-SDT |
|---|---|---|---|---|---|---|
| CP | 0.000 | 0.000 | 0.000 | **0.632** | **1.214** | **4.075** |
| AB | 0.035 | -0.630 | 0.104 | **0.728** | 0.338 | **0.982** |
| LL | 0.341 | **0.750** | 0.370 | **4.776** | **8.155** | **2.254** |
| MC-C | -0.042 | -0.002 | -1.035 | -2.011 | -1.172 | **0.745** |
| PD-C | **1.195** | 0.081 | 0.468 | **13.120** | **4.101** | **7.573** |
| Mean ↓ | **0.306** | **0.040** | **-0.019** | **3.449** | **2.527** | **3.126** |

## A.3 COMPARISON OF SYMPOL, SA-DT (DAGGER) AND VIPER (Q-DAGGER)

In this section, we provide a direct comparison of SYMPOL with SA-DT and VIPER for control environments (see Table 8) and MiniGrid environments (see Table 9). SA-DT (Silva et al., 2020) can be considered as a version of DAGGER (Ross et al., 2011) and is conceptually similar to VIPER (Q-DAGGER) (Bastani et al., 2018) which improves data collection including additional weighting. Our results remain consistent with our original claims, demonstrating that SYMPOL outperforms alternative approaches. This is also in-line with the results reported by Kohler et al. (2024), where the authors show, that the sampling in VIPER does not yield a better performance compared to DAGGER/SA-DT for interpretable DTs. The results reported in the original VIPER paper (Bastani et al., 2018) stating to achieve a perfect reward for CP are on a different version of the environment (CartPole-v0) with only 200 opposed to 500 time steps and less randomness (CartPole-v1), making the underlying task easier. Also, we want to note, that the reported results are in-line with related work, reporting comparable or worse results than ours. For instance, Vos & Verwer (2024) report a mean reward of only 367 for VIPER on CartPole-v1. Also, Kenny et al. (2023) showcase a poor

Table 8: **Control Performance.** We report the average undiscounted cumulative test reward over 25 random trials. The best interpretable method, and methods not statistically different, are marked bold. Please note that VIPER cannot be applied to continuous environments.

|             | CP  | AB    | LL    | MC-C | PD-C  |
|-------------|-----|-------|-------|------|-------|
| SYMPOL (ours) | **500** | **- 80** | **- 57** | **94** | **- 323** |
| D-SDT       | 128 | -205  | -221  | -10  | -1343 |
| SA-DT (d=5) | 446 | -97   | -197  | **97** | -1251 |
| SA-DT (d=8) | 476 | **- 75** | -150  | **96** | **- 854** |
| VIPER (d=5) | 457 | **- 77** | -200  | -    | -     |
| VIPER (d=8) | 480 | **- 75** | -169  | -    | -     |
| MLP         | 500 | - 72  | 241   | 95   | - 191 |
| SDT         | 500 | - 77  | -124  | - 4  | - 310 |

Table 9: **MiniGrid Performance.** We report the average undiscounted cumulative test reward over 25 random trials. The best interpretable method, and methods not statistically different, are marked bold.

|             | E-R   | DK      | LG-5    | LG-7    | DS      |
|-------------|-------|---------|---------|---------|---------|
| SYMPOL (ours) | **0.964** | **0.959** | **0.951** | **0.953** | 0.939   |
| D-SDT       | 0.662 | 0.654   | 0.262   | 0.381   | 0.932   |
| SA-DT (d=5) | 0.583 | **0.958** | **0.951** | 0.458   | **0.952** |
| SA-DT (d=8) | 0.845 | **0.961** | **0.951** | 0.799   | **0.954** |
| VIPER (d=5) | 0.651 | **0.958** | **0.948** | 0.456   | **0.954** |
| VIPER (d=8) | 0.845 | **0.963** | **0.948** | 0.801   | **0.954** |
| MLP         | 0.963 | 0.963   | 0.951   | 0.760   | 0.951   |
| SDT         | 0.966 | 0.959   | 0.839   | 0.953   | 0.954   |

performance of VIPER in general and specifically for LL the performance is worse than what we reported. Our findings align with these, suggesting that differences in performance may reflect randomness and missing generalizability in the evaluation.

## A.4 TREE SIZE

We report the average tree sizes over 25 trials for each environment. The DTs for SYMPOL and D-SDT are automatically pruned by removing redundant paths. There are mainly two identifiers, making a path redundant:

- The split threshold of a split is outside the range specified by the environment. For instance, if $x_1 \in [0.0, 1.0]$ the decision $x_1 \leq -0.1$ will always be false as $-0.1 \leq 0.0$.

- A decision at a higher level of the tree already predefines the current decision. For instance, if the split at the root node is $x_1 \leq 0.5$ and the subsequent node following the true path is $x_1 \leq 0.6$ we know that this node will always be evaluated to true as $0.5 \leq 0.6$.

We excluded the MiniGrid environments here, as they require a more sophisticated, automated pruning as there exist more requirements making a path redundant. For instance, if for the decision whether there is a wall in front of the agent is true, the decision for all other objects at the same position has to be always false.

Table 10: **Tree Size**. We report the average size of the learned DT for each environment.

|  | SYMPOL (ours) | D-SDT | SA-DT (d=5) | SA-DT (d=8) |
|---|---|---|---|---|
| CP | 39.4 | 14.2 | 61.8 | 315.0 |
| AB | 78.6 | 17.0 | 56.5 | 173.0 |
| LL | 55.0 | 19.8 | 59.8 | 270.2 |
| MC-C | 23.4 | 3.0 | 61.0 | 311.8 |
| PD-C | 56.2 | 28.6 | 62.2 | 388.2 |
| **Mean** $\downarrow$ | **50.5** | **16.5** | **60.3** | **291.6** |

## A.5 INTERPRETABILITY COMPARISON BETWEEN HARD, AXIS-ALIGNED AND SOFT DTs

In the main paper, we showed a visualization of a hard, axis-aligned DT learned by SYMPOL on the MC-C environment. While this was a comparatively small tree, we provide another example of a comparatively large tree with 59 nodes in Figure 10. While the tree is comparatively large, we can observe that the main logic is contained in the nodes at higher levels, focusing on the pole angle and the pole angular velocity. The less important features are in the lower levels where splits are often mage on the cart position, which is not required to solve the task perfectly. This also highlights the potential for advanced post-hoc pruning methods to increase interpretability and potentially even generalization. When comparing the hard, axis-aligned decision trees (DTs) (Figure 10 and Figure 4) learned by SYMPOL with corresponding soft decision trees (SDTs) learned by existing direct optimization methods (Figure 11 and Figure 12), the superiority of axis-aligned splits over oblique, multidimensional boundaries becomes evident.

The DTs learned by SYMPOL are substantially more interpretable, both at the level of individual splits and the overall tree structure. For example, when examining the root node of the CartPole task, the standard DT (Figure 10) makes a straightforward comparison, "$s_3$(Pole Velocity) $\leq -0.800$", whereas the corresponding SDT (Figure 11) expresses the decision as "$\sigma(-1.14s_0 - 0.30s_1 + 0.94s_2 + 0.11s_3 + 0.99)$", which is significantly more complex and harder to interpret. This disparity becomes even more pronounced when considering complete paths or the entire tree. In SYMPOL's DTs, decisions at nodes are binary (yes/no), whereas SDTs employ probabilistic routing.

Probabilistic routing introduces two key disadvantages in interpretability compared to axis-aligned DTs: (1) the need to consider multiple paths simultaneously, and (2) the inability to directly interpret the leaf outputs, as they are weighted by path probabilities. We also want to note that in order to allow a visualization of the SDT, we reduced the tree size to only 4, while the tree size used during the evaluation was 7.

## A.6 ABLATION STUDY

Our ablation study was designed to support our intuitive justifications for the modifications made to the RL framework and our method. Therefore, we disabled individual components of our method and evaluated the performance without the specific component. This includes the following modifications introduced in Section 4:

1. **w/o separate architecture:** Instead of using separate architectures for actor and critic, we use the same architecture and hyperparameters for the actor and critic.

2. **w/o dynamic rollout:** We proposed a dynamic rollout buffer that increases with a stepwise exponential rate during training to increase stability while maintaining exploration early on. Here we used a standard, static rollout buffer.

3. **w/o batch size adjustment:** Similar to the dynamic rollout buffer, we proposed using a dynamic batch size to increase gradient stability in later stages of the training. Here, we used standard, static batch size.

4. **w/o adamW:** We introduced an Adam optimizer with weight decay to SYMPOL to support the adjustment of the features to split on and the class predicted. Here, we use a standard Adam optimizer without weight decay.

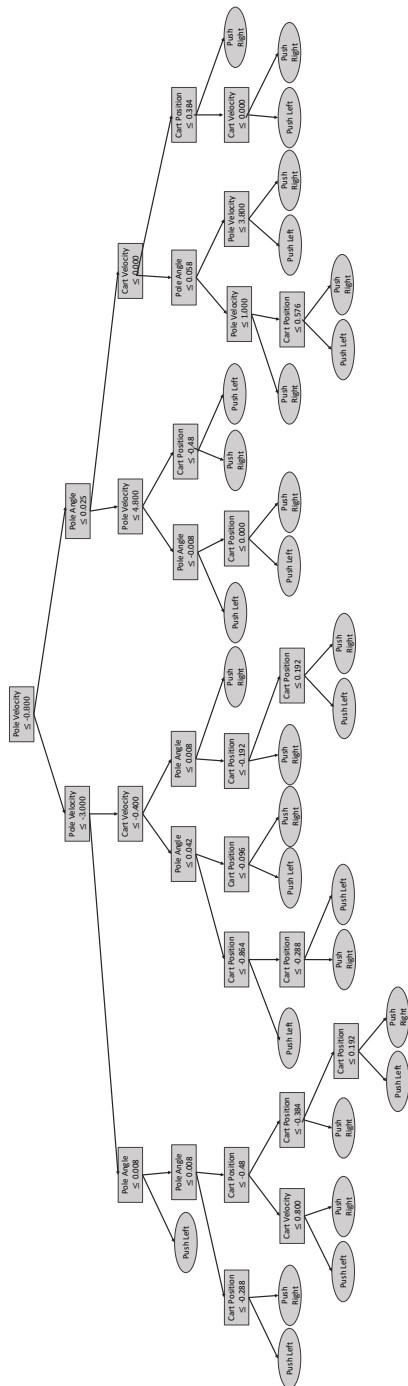

Figure 10: **Exemplary large CartPole DT.** This figure visualizes a larger tree with 59 nodes learned by SYMPOL on the CartPole environment.

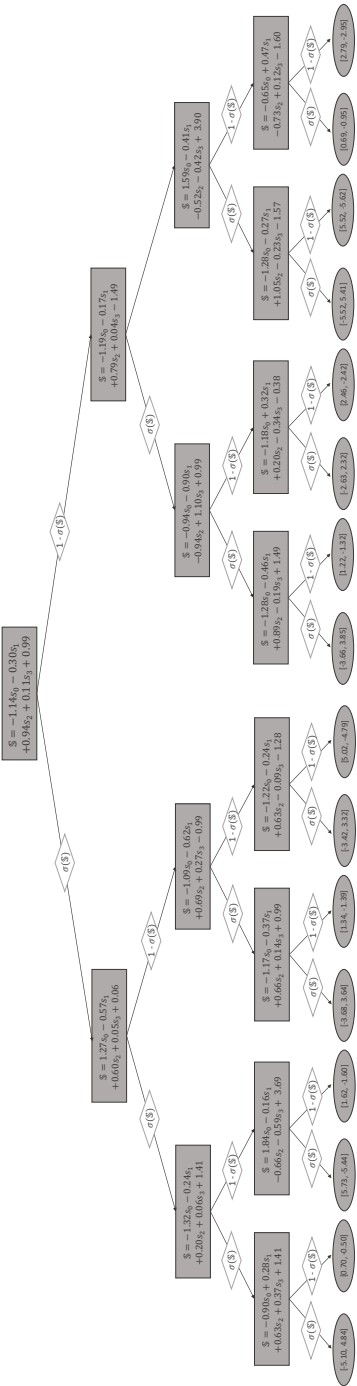

Figure 11: **Exemplary SDT for CartPole.** This figure visualizes a soft / differentiable DT learned on CartPole. The tree involves oblique decisions involving multiple features at each split. Additionally, there is no hard decision on which path is selected, but multiple paths are taken with an associated probability. The final prediction is obtained by weighting the leaf outputs with the corresponding path probability.

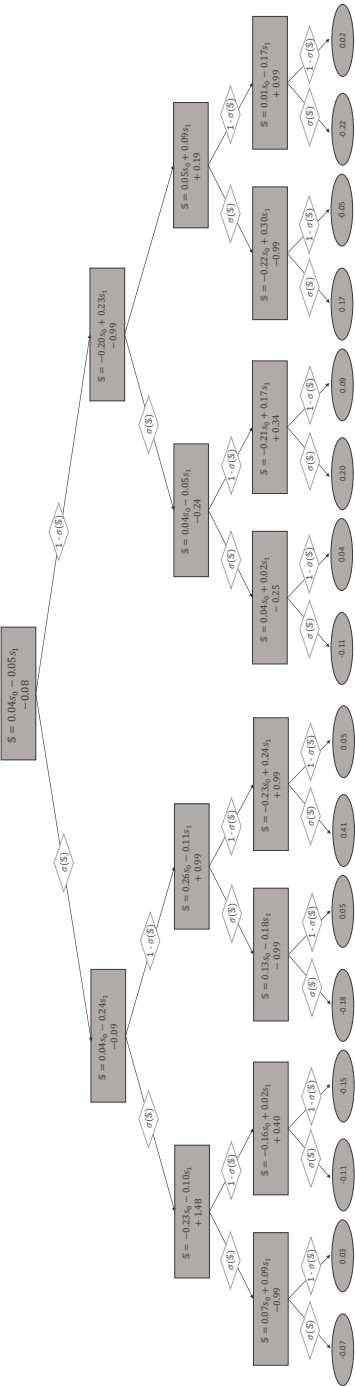

Figure 12: **Exemplary SDT for MountainCar.** This figure visualizes a soft / differentiable DT learned on MountainCar. The tree involves oblique decisions involving multiple features at each split. Additionally, there is no hard decision on which path is selected, but multiple paths are taken with an associated probability. The final prediction is obtained by weighting the leaf outputs with the corresponding path probability.

Detailed results for each of the control datasets are reported in Table 11. The results clearly confirm our intuitive justifications, as each adjustment has a crucial impact on the performance of SYMPOL.

Table 11: **Ablation Study**. We report the average test performance over a total of 25 random trials. This normalized performance consists in normalizing each reward between 0 and 1 via an affine renormalization between the top- and worse-performing models. Instead of the worse-performing model, we use the 20% test reward quantile to account for outliers.

| Agent Type | CP | AB | LL | MC-C | PD-C | Normalized Mean ($\uparrow$) |
|---|---|---|---|---|---|---|
| SYMPOL | **500.0** | **- 79.9** | **- 57.4** | **94.3** | **- 323.3** | **0.988** |
| w/o separate architecture | 135.6 | -196.4 | -276.8 | -552.4 | -1219.4 | 0.080 |
| w/o dynamic rollout | 456.1 | - 92.0 | -178.2 | -144.1 | - 434.4 | 0.598 |
| w/o batch size adjustment | 498.8 | - 81.7 | -320.7 | -1818.8 | - 434.4 | 0.372 |
| w/o adamW | 416.1 | - 78.3 | - 97.3 | 0.0 | - 393.5 | 0.865 |

## A.7 RUNTIME AND TRAINING CURVES

The experiments were conducted on a single NVIDIA RTX A6000. The environments for CP, AB, MC-C and PD-C were vectorized (Lange, 2022) and therefore the training is highly efficient, taking only 30 seconds for 1mio timesteps on average (excluding the sequential evaluation which cannot be vectorized). The remaining environments are not vectorized, and we used the standard Gymnasium (Towers et al., 2024) implementation. In Table 12 can clearly see the impact of environment vectorization, as the runtime for LL, which is not vectorized, is more than 10 times higher with over 400 seconds.

Table 12: **Runtime.** We report the average runtime over 25 trials. One trial spans 1mio timesteps for each environment. We excluded LL from the mean runtime calculation, as this is the only non-vectorized environment. To provide a fair comparison of different methods, we aligned the step and batch size.

| | SYMPOL (ours) | SDT | MLP |
|---|---|---|---|
| CP | 28.8 | 23.9 | 25.2 |
| AB | 35.5 | 37.7 | 33.8 |
| MC-C | 23.4 | 19.4 | 18.4 |
| PD-C | 28.7 | 28.2 | 18.5 |
| **Mean** $\downarrow$ | **29.1** | **27.3** | **24.0** |
| LL | 402.3 | 394.0 | 405.6 |

In addition to the training times, we report detailed training curves for each method. Figure 13 compares the training reward and the test reward of SYMPOL with the full-complexity models MLP and SDT. SYMPOL shows a similar convergence compared to full-complexity models on most environments. For AB, SYMPOL converges even faster than an MLP which can be attributed to the dynamic rollout buffer and batch size. For MC-C we can see that the training of SYMPOL is very unstable at the beginning. We believe that this can be attributed to the sparse reward function of this certain environment and the fact that as a result, minor changes in the policy can result in a severe drop in the reward. Combined with the small rollout buffer and batch size early in the training of SYMPOL, this can result in an unstable training. However, we can see that the training stabilizes later on, which again confirms the design of our dynamic buffer size increasing over time.

Furthermore, we provide a pairwise comparison of SYMPOL with SA-DT and D-SDT in Figure 14. Here, we can again observe the severe information loss for D-SDT and SA-DT by comparing the training curve with the test reward.

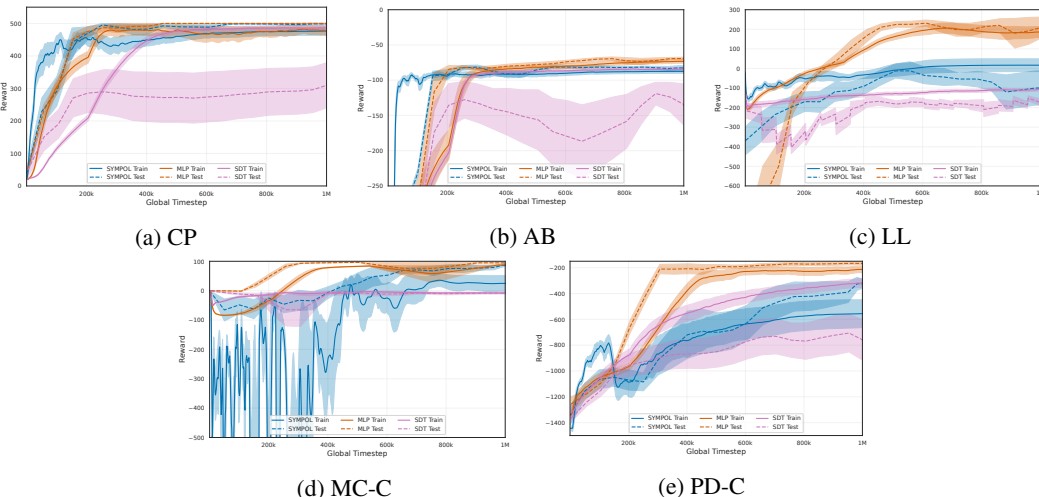

(a) CP          (b) AB          (c) LL

(d) MC-C          (e) PD-C

Figure 13: **Training Curves (Full-Complexity).** Shows the training reward as solid line and the test reward as dashed line for SYMPOL (blue), MLP (orange) and SDT (green).

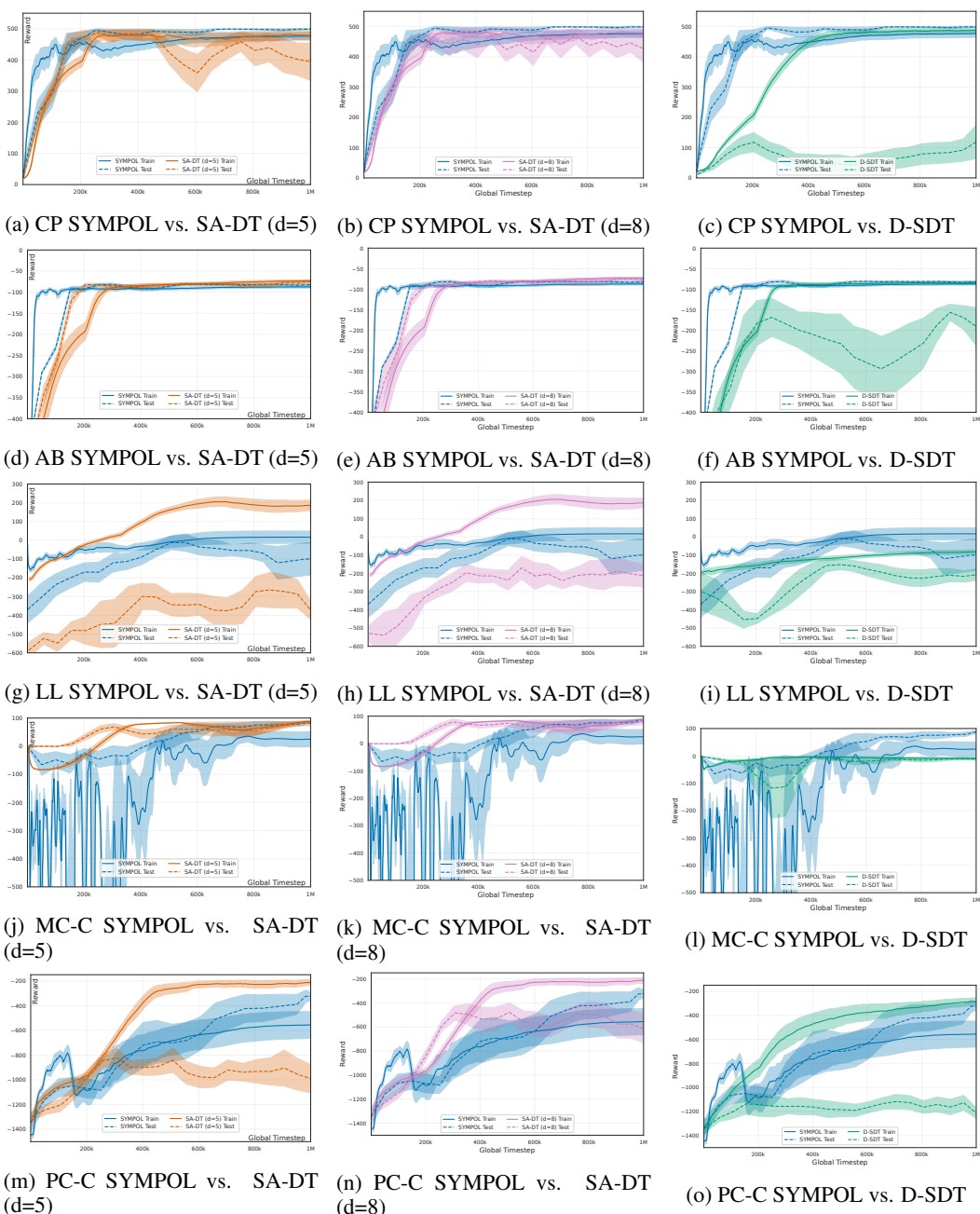

(a) CP SYMPOL vs. SA-DT (d=5) (b) CP SYMPOL vs. SA-DT (d=8) (c) CP SYMPOL vs. D-SDT

(d) AB SYMPOL vs. SA-DT (d=5) (e) AB SYMPOL vs. SA-DT (d=8) (f) AB SYMPOL vs. D-SDT

(g) LL SYMPOL vs. SA-DT (d=5) (h) LL SYMPOL vs. SA-DT (d=8) (i) LL SYMPOL vs. D-SDT

(j) MC-C SYMPOL vs. SA-DT (d=5) (k) MC-C SYMPOL vs. SA-DT (d=8) (l) MC-C SYMPOL vs. D-SDT

(m) PC-C SYMPOL vs. SA-DT (d=5) (n) PC-C SYMPOL vs. SA-DT (d=8) (o) PC-C SYMPOL vs. D-SDT

Figure 14: **Training Curves.** Shows the training reward as solid line and the test reward as dashed line for SYMPOL (blue), SA-DT-5 (orange) SA-DT-8 (green) and D-SDT (red). Thereby, the test reward is calculated with the discretized/distilled policy for SA-DT and D-SDT. For several datasets, we can again observe the severe information loss introduced with the post-processing (e.g. for PD-C and LL).

# B MINIGRID

We used the MiniGrid implementation from (Chevalier-Boisvert et al., 2023). For each environment, we limited to observations and the actions to the available ones according to the documentation. Furthermore, we decided to use a view size of 3 to allow a good visualization of the results. In the following, we provide more examples for our MiniGrid Use-Case, along with more detailed visual-

izations. In the following, we visualized the SYMPOL agent sequentially acting in the environment as one image for one step from left to right and top to bottom. Figure 15 shows how SYMPOL (see image in the main paper or `tree_function(obs)` defined below) solves the environment. Figure 16 and Figure 18 show the same agent failing on the environment with domain randomization, proving that the agent did not generalize, as we could already observe by inspecting the symbolic, tree-based policy. Retraining the agent with domain randomization (see image in the main paper or `tree_function_retrained(obs)` defined below), SYMPOL is able to solve the environment (see Figure 17 and Figure 19), maintaining interpretability.

## B.1 VISUALIZATIONS ENVIRONMENT

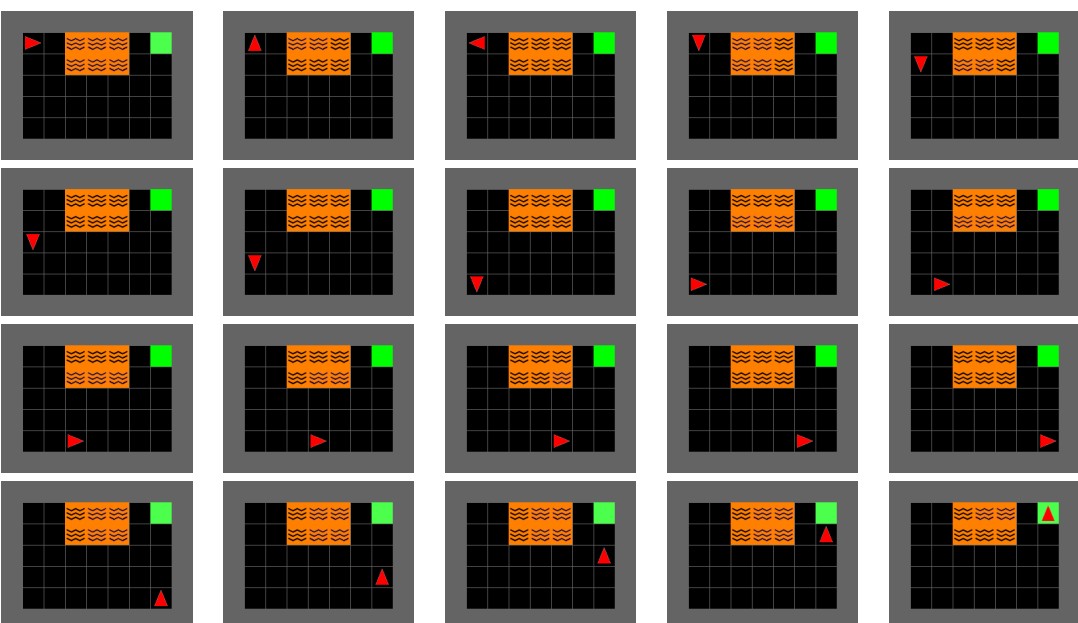

Figure 15: **DistShift SYMPOL.** This figure visualizes the path taken by the SYMPOL agent trained on the basic DistShift environment (see image in the main paper or `tree_function(obs)` defined below) from left to right and top to bottom. The agent follows the wall and reaches the goal at the top right corner.

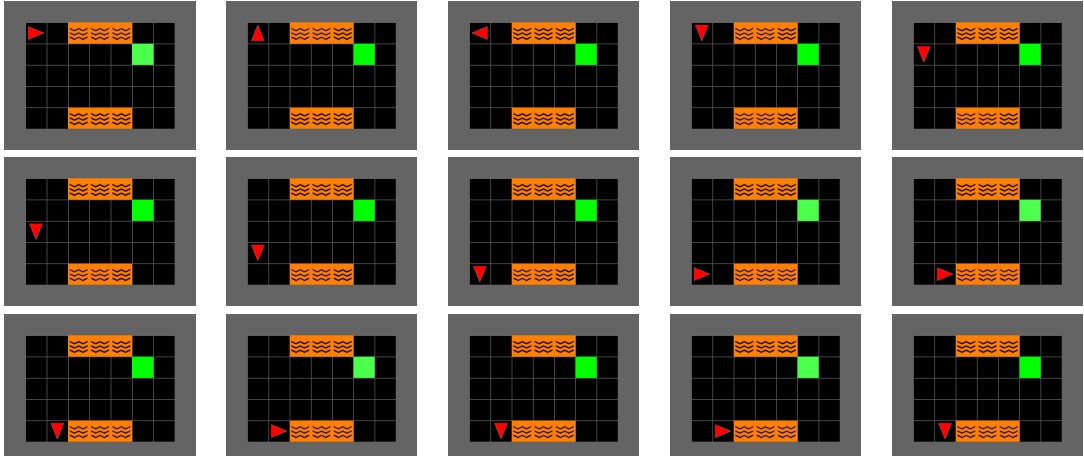

Figure 16: **DistShift (Domain Randomization) SYMPOL Example 1.** This figure visualizes the path taken by the SYMPOL agent trained on the basic DistShift environment (see image in the main paper or `tree_function(obs)` defined below) from left to right and top to bottom. The agent follows the wall gets stuck by the lava.

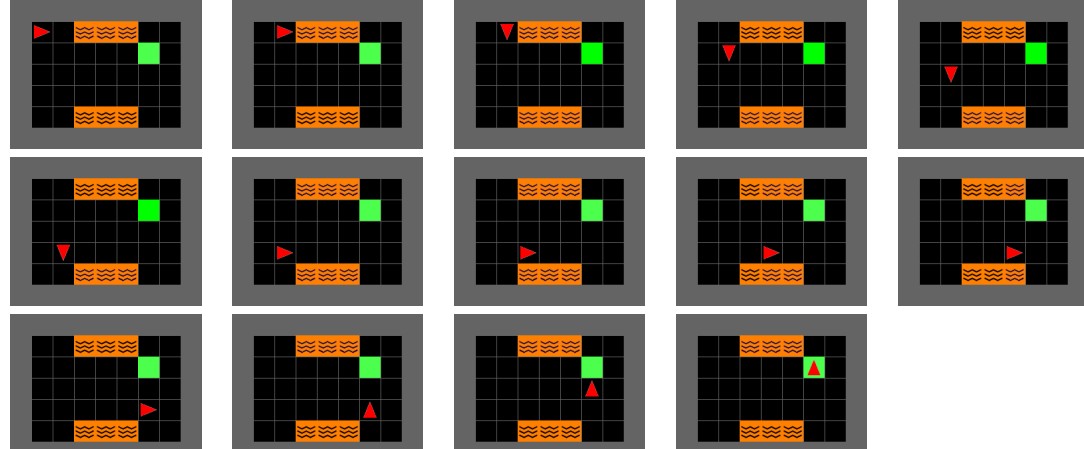

Figure 17: **DistShift (Domain Randomization) SYMPOL (retrained) Example 1.** This figure visualizes the path taken by the SYMPOL agent trained on the randomized DistShift environment (see image in the main paper or `tree_function_retrained(obs)` defined below) from left to right and top to bottom. The agent avoids the lava, identifies the goal and walks into the goal.

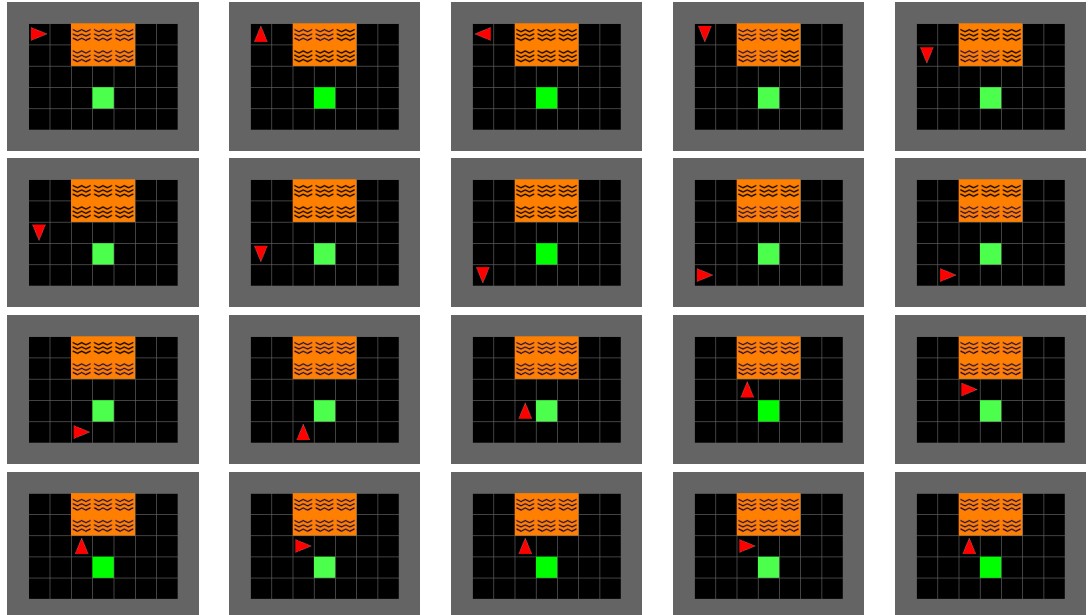

Figure 18: **DistShift (Domain Randomization) SYMPOL Example 2.** This figure visualizes the path taken by the SYMPOL agent trained on the basic DistShift environment (see image in the main paper or `tree_function(obs)` defined below) from left to right and top to bottom. The agent follows the wall until there is no empty space on the left. Instead of an empty space there is the goal, but instead of walking into the goal, the agent surpasses it and again gets stuck at the lava.

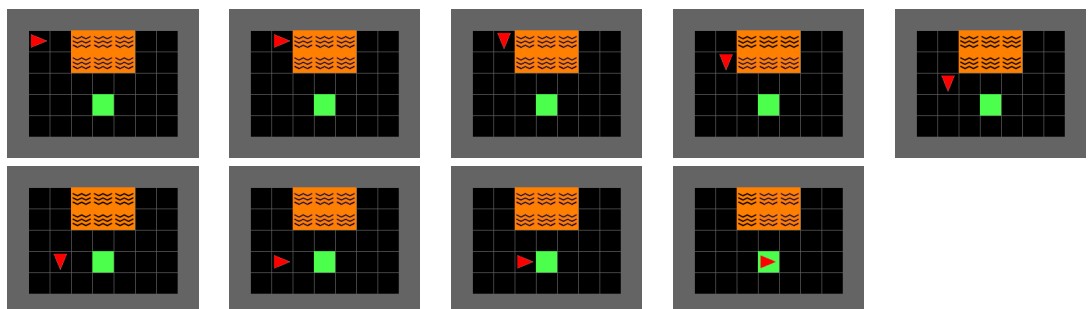

Figure 19: **DistShift (Domain Randomization) SYMPOL (retrained) Example 2.** This figure visualizes the path taken by the SYMPOL agent trained on the randomized DistShift environment (see image in the main paper or `tree_function_retrained(obs)` defined below) from left to right and top to bottom. The agent avoids the lava, identifies the goal, and walks into the goal.

### B.2 SYMPOL ALGORITHMIC PRESENTATION

```python
def tree_function(obs):
    if obs[field one to front and one to left] is 'empty':
        if obs[field one to front] is 'lava':
            if obs[field one to left] is 'empty':
                action = 'turn right'
            else:
                action = 'turn left'
        else:
            action = 'move forward'
    else:
        action = 'turn left'
    return action
```

```python
def tree_function_retrained(obs):
    if obs[field one to left] is 'goal':
        action = 'turn left'
    else:
        if obs[field one to right] is 'goal':
            action = 'turn right'
        else:
            if obs[field two to front] is 'wall':
                if obs[field one to front and one to right] is 'goal':
                    if obs[field one to right] is 'lava':
                        action = 'turn left'
                    else:
                        action = 'move forward'
                else:
                    if obs[field one to front] is 'goal':
                        action = 'move forward'
                    else:
                        action = 'turn left'
            else:
                if obs[field one to front] is 'lava':
                    action = 'turn right'
                else:
                    action = 'move forward'
    return action
```

## C  METHODS AND HYPERPARAMETERS

The main methods we compared SYMPOL against are behavioral cloning state-action DTs (SA-DT) and discretized soft decision trees (D-SDT). In addition to the information given in the paper, we want to provide some more detailed results of the implementation and refer to our source code for the exact definition.

- **State-Action DTs (SA-DT)** Behavioral cloning SA-DTs are the most common method to generate interpretable policies post-hoc. Hereby, we first train an MLP policy, which is then distilled into a DT as a post-processing step after the training. Specifically, we train the DT on a dataset of expert trajectories generated with the MLP policy. The number of expert trajectories was set to 25 which we experienced as a good trade-off between dataset size for the distillation and model complexity during preliminary experiments. The 25 expert trajectories result in a total of approximately 12500 state-action pairs, varying based on the environment specification.

- **Discretized Soft Decision Trees (D-SDT)** SDTs allow gradient computation by assigning probabilities to each node. While SDTs exhibit a hierarchical structure, they are usually considered as less interpretable, since multiple features are considered in a single split and the whole tree is traversed simultaneously (Marton et al., 2024a). Therefore, Silva et al.

(2020) use SDTs as policies which are discretized post-hoc to allow an easy interpretation considering only a single feature at each split. Discretization is achieved by employing an argmax to obtain the feature index and normalizing the split threshold based on the feature vector. We improved their method by replacing the scaled sigmoid and softmax, with an entmoid and entmax transformation(Peters et al., 2019), resulting in sparse feature selectors with more responsive gradients, as it is common practice Popov et al. (2019); Chang et al. (2021).

In the following, we list the parameter grids used during the hyperparameter optimization (HPO) as well as the optimal parameters selected for each environment. For SYMPOL, SDT and MLP, we optimized the hyperparameters based on the validation reward with optuna Akiba et al., 2019 for 60 trials. Thereby, we ensured that the environments evaluated during the HPO were distinct to the environments used for reporting the test performance in the rest of the paper. Additionally, we decrease the learning rate if no improvement in validation reward is observed for five consecutive iterations, allowing for finer model adjustments in later training stages.

## C.1 HPO GRIDS

Table 13: **HPO Grid SYMPOL**

| hyperparameter | values |
| --- | --- |
| learning_rate_actor_weights | [0.0001, 0.1] |
| learning_rate_actor_split_values | [0.0001, 0.05] |
| learning_rate_actor_split_idx_array | [0.0001, 0.1] |
| learning_rate_actor_leaf_array | [0.0001, 0.05] |
| learning_rate_actor_log_std | [0.0001, 0.1] |
| learning_rate_critic | [0.0001, 0.01] |
| n_update_epochs | [0, 10] |
| reduce_lr | {True, False} |
| n_steps | {128, 512} |
| n_envs | [4, 16] |
| norm_adv | {True, False} |
| ent_coef | {0.0, 0.1, 0.2, 0.5} |
| gae_lambda | {0.8, 0.9, 0.95, 0.99} |
| gamma | {0.9, 0.95, 0.99, 0.999} |
| vf_coef | {0.25, 0.50, 0.75} |
| max_grad_norm | [None] |
| SWA | {True} |
| adamW | {True} |
| depth | {7} |
| minibatch_size | {64} |

Table 14: **HPO Grid MLP**

| hyperparameter | values |
|---|---:|
| neurons_per_layer | [16, 256] |
| num_layers | [1, 3] |
| learning_rate_actor | [0.0001, 0.01] |
| learning_rate_critic | [0.0001, 0.01] |
| minibatch_size | {64, 128, 256, 512} |
| n_update_epochs | [1, 10] |
| n_steps | {128, 512} |
| n_envs | [4, 16] |
| norm_adv | {True, False} |
| ent_coef | {0.0, 0.1, 0.2, 0.5} |
| gae_lambda | {0.8, 0.9, 0.95, 0.99} |
| gamma | {0.9, 0.95, 0.99, 0.999} |
| vf_coef | {0.25, 0.50, 0.75} |
| max_grad_norm | {0.1, 0.5, 1.0, None} |

Table 15: **HPO Grid SDT**

| hyperparameter | values |
|---|---:|
| critic | {'MLP', 'SDT'} |
| depth | [4, 8] |
| temperature | {0.01, 0.05, 0.1, 0.5, 1.0} |
| learning_rate_actor | [0.0001, 0.01] |
| learning_rate_critic | [0.0001, 0.01] |
| minibatch_size | {64, 128, 256, 512} |
| n_update_epochs | [1, 10] |
| n_steps | {128, 512} |
| n_envs | [4, 16] |
| norm_adv | {True, False} |
| ent_coef | {0.0, 0.1, 0.2, 0.5} |
| gae_lambda | {0.8, 0.9, 0.95, 0.99} |
| gamma | {0.9, 0.95, 0.99, 0.999} |
| vf_coef | {0.25, 0.50, 0.75} |
| max_grad_norm | {0.1, 0.5, 1.0, None} |

## C.2 BEST HYPERPARAMETERS

Table 16: **Best Hyperparameters SYMPOL (Control)**

|  | CP | AB | LL | MC-C | PD-C |
|---|---|---|---|---|---|
| ent_coef | 0.200 | 0.000 | 0.000 | 0.500 | 0.100 |
| gae_lambda | 0.950 | 0.950 | 0.900 | 0.990 | 0.800 |
| gamma | 0.990 | 0.990 | 0.999 | 0.999 | 0.999 |
| learning_rate_actor_weights | 0.048 | 0.003 | 0.072 | 0.000 | 0.022 |
| learning_rate_actor_split_values | 0.000 | 0.000 | 0.001 | 0.000 | 0.000 |
| learning_rate_actor_split_idx_array | 0.026 | 0.052 | 0.010 | 0.000 | 0.010 |
| learning_rate_actor_leaf_array | 0.020 | 0.005 | 0.009 | 0.028 | 0.006 |
| learning_rate_actor_log_std | 0.001 | 0.002 | 0.021 | 0.094 | 0.000 |
| learning_rate_critic | 0.001 | 0.000 | 0.002 | 0.002 | 0.000 |
| max_grad_norm | None | None | None | None | None |
| n_envs | 7 | 8 | 6 | 5 | 15 |
| n_steps | 512 | 128 | 512 | 128 | 128 |
| n_update_epochs | 7 | 7 | 7 | 2 | 7 |
| norm_adv | False | False | True | False | True |
| reduce_lr | True | True | True | True | False |
| vf_coef | 0.500 | 0.250 | 0.500 | 0.500 | 0.750 |
| SWA | True | True | True | True | True |
| adamW | True | True | True | True | True |
| dropout | 0.000 | 0.000 | 0.000 | 0.000 | 0.000 |
| depth | 7 | 7 | 7 | 7 | 7 |
| minibatch_size | 64 | 64 | 64 | 64 | 64 |
| n_estimators | 1 | 1 | 1 | 1 | 1 |

Table 17: **Best Hyperparameters SYMPOL (MiniGrid)**

|  | E-R | DK | LG-5 | LG-7 | DS |
|---|---|---|---|---|---|
| ent_coef | 0.100 | 0.200 | 0.100 | 0.100 | 0.500 |
| gae_lambda | 0.990 | 0.950 | 0.900 | 0.900 | 0.950 |
| gamma | 0.900 | 0.990 | 0.950 | 0.990 | 0.999 |
| learning_rate_actor_weights | 0.063 | 0.042 | 0.055 | 0.001 | 0.036 |
| learning_rate_actor_split_values | 0.001 | 0.001 | 0.006 | 0.001 | 0.000 |
| learning_rate_actor_split_idx_array | 0.001 | 0.001 | 0.012 | 0.001 | 0.009 |
| learning_rate_actor_leaf_array | 0.003 | 0.004 | 0.009 | 0.008 | 0.001 |
| learning_rate_actor_log_std | 0.043 | 0.021 | 0.005 | 0.002 | 0.038 |
| learning_rate_critic | 0.001 | 0.001 | 0.001 | 0.001 | 0.001 |
| max_grad_norm | None | None | None | None | None |
| n_envs | 14 | 14 | 16 | 7 | 10 |
| n_steps | 128 | 512 | 512 | 128 | 512 |
| n_update_epochs | 8 | 9 | 5 | 4 | 5 |
| norm_adv | True | True | True | True | False |
| reduce_lr | False | True | True | True | True |
| vf_coef | 0.500 | 0.500 | 0.250 | 0.500 | 0.250 |
| SWA | True | True | True | True | True |
| adamW | True | True | True | True | True |
| dropout | 0.000 | 0.000 | 0.000 | 0.000 | 0.000 |
| depth | 7 | 7 | 7 | 7 | 7 |
| minibatch_size | 64 | 64 | 64 | 64 | 64 |
| n_estimators | 1 | 1 | 1 | 1 | 1 |

Table 18: **Best Hyperparameters MLP (Control)**

|                       | CP    | AB    | LL    | MC-C  | PD-C  |
| --------------------- | ----- | ----- | ----- | ----- | ----- |
| adamW                 | False | False | False | False | False |
| ent_coef              | 0.200 | 0.000 | 0.100 | 0.100 | 0.100 |
| gae_lambda            | 0.900 | 0.900 | 0.900 | 0.950 | 0.950 |
| gamma                 | 0.999 | 0.990 | 0.999 | 0.999 | 0.990 |
| learning_rate_actor   | 0.001 | 0.000 | 0.001 | 0.005 | 0.000 |
| learning_rate_critic  | 0.003 | 0.005 | 0.003 | 0.001 | 0.002 |
| max_grad_norm         | 1.000 | 1.000 | 0.500 | 0.100 | None  |
| minibatch_size        | 256   | 256   | 128   | 512   | 128   |
| n_envs                | 13    | 12    | 13    | 15    | 8     |
| n_steps               | 128   | 512   | 512   | 512   | 512   |
| n_update_epochs       | 7     | 9     | 8     | 2     | 2     |
| neurons_per_layer     | 139   | 185   | 46    | 240   | 75    |
| norm_adv              | False | True  | False | True  | True  |
| num_layers            | 2     | 2     | 3     | 2     | 2     |
| reduce_lr             | False | False | False | False | False |
| vf_coef               | 0.250 | 0.500 | 0.500 | 0.250 | 0.250 |

Table 19: **Best Hyperparameters MLP (MiniGrid)**

|                       | E-R   | DK    | LG-5  | LG-7  | DS    |
| --------------------- | ----- | ----- | ----- | ----- | ----- |
| adamW                 | False | False | False | False | False |
| ent_coef              | 0.100 | 0.100 | 0.100 | 0.100 | 0.100 |
| gae_lambda            | 0.950 | 0.900 | 0.950 | 0.950 | 0.990 |
| gamma                 | 0.990 | 0.900 | 0.990 | 0.900 | 0.990 |
| learning_rate_actor   | 0.000 | 0.000 | 0.002 | 0.000 | 0.000 |
| learning_rate_critic  | 0.001 | 0.000 | 0.003 | 0.001 | 0.001 |
| max_grad_norm         | 0.100 | 0.100 | 1     | 0.500 | 0.100 |
| minibatch_size        | 64    | 256   | 128   | 512   | 256   |
| n_envs                | 13    | 8     | 8     | 12    | 10    |
| n_steps               | 512   | 256   | 512   | 128   | 128   |
| n_update_epochs       | 5     | 7     | 9     | 8     | 7     |
| neurons_per_layer     | 112   | 169   | 76    | 28    | 158   |
| norm_adv              | False | True  | False | True  | True  |
| num_layers            | 3     | 1     | 1     | 1     | 2     |
| reduce_lr             | False | False | False | False | False |
| vf_coef               | 0.500 | 0.500 | 0.250 | 0.750 | 0.500 |

Table 20: **Best Hyperparameters SDT (Control)**

|                     | CP    | AB    | LL    | MC-C  | PD-C  |
|---------------------|-------|-------|-------|-------|-------|
| adamW               | False | False | False | False | False |
| critic              | mlp   | mlp   | mlp   | mlp   | mlp   |
| depth               | 7     | 6     | 8     | 7     | 7     |
| ent_coef            | 0.000 | 0.100 | 0.200 | 0.000 | 0.200 |
| gae_lambda          | 0.950 | 0.950 | 0.990 | 0.900 | 0.900 |
| gamma               | 0.990 | 0.990 | 0.999 | 0.990 | 0.900 |
| learning_rate_actor | 0.001 | 0.002 | 0.001 | 0.001 | 0.000 |
| learning_rate_critic| 0.000 | 0.000 | 0.001 | 0.007 | 0.000 |
| max_grad_norm       | 0.100 | 0.100 | 1.000 | 0.500 | 0.100 |
| minibatch_size      | 128   | 128   | 128   | 64    | 128   |
| n_envs              | 15    | 6     | 7     | 14    | 7     |
| n_steps             | 512   | 128   | 512   | 512   | 256   |
| n_update_epochs     | 4     | 10    | 2     | 1     | 7     |
| norm_adv            | True  | False | True  | False | False |
| reduce_lr           | False | False | False | False | False |
| temperature         | 1     | 0.500 | 1     | 1     | 0.100 |
| vf_coef             | 0.500 | 0.500 | 0.750 | 0.250 | 0.500 |

Table 21: **Best Hyperparameters SDT (MiniGrid)**

|                     | E-R   | DK    | LG-5  | LG-7  | DS    |
|---------------------|-------|-------|-------|-------|-------|
| adamW               | False | False | False | False | False |
| critic              | sdt   | mlp   | sdt   | sdt   | sdt   |
| depth               | 7     | 6     | 7     | 8     | 7     |
| ent_coef            | 0.100 | 0.100 | 0.200 | 0.100 | 0.100 |
| gae_lambda          | 0.900 | 0.950 | 0.990 | 0.950 | 0.900 |
| gamma               | 0.990 | 0.900 | 0.999 | 0.950 | 0.950 |
| learning_rate_actor | 0.004 | 0.001 | 0.000 | 0.002 | 0.001 |
| learning_rate_critic| 0.000 | 0.002 | 0.000 | 0.005 | 0.002 |
| max_grad_norm       | 0.100 | 0.100 | 0.500 | 0.100 | None  |
| minibatch_size      | 512   | 256   | 512   | 256   | 512   |
| n_envs              | 10    | 10    | 10    | 13    | 5     |
| n_steps             | 512   | 256   | 256   | 128   | 512   |
| n_update_epochs     | 5     | 10    | 8     | 4     | 7     |
| norm_adv            | True  | True  | True  | True  | True  |
| reduce_lr           | False | False | False | False | False |
| temperature         | 1     | 1     | 1     | 1     | 1     |
| vf_coef             | 0.750 | 0.750 | 0.750 | 0.250 | 0.750 |

