# OpenReview forum: "Mitigating Information Loss in Tree-Based Reinforcement Learning via Direct Optimization"
_ICLR.cc/2025/Conference — ICLR 2025 Spotlight_

### Official Review · Reviewer_6tAh · 2024-10-26

**Soundness:** 4
**Presentation:** 4
**Contribution:** 3
**Rating:** 8
**Confidence:** 5

**Summary:**

The paper presents SYMPOL, a new approach to learning interpretable policies for reinforcement learning problems by building on recent works facilitating gradient-based training of decision tree architectures. SYMPOL performs much better than prior methods for producing decision tree policies (such as discretizing policies or behavior-cloning a decision tree from a replay buffer), and includes a few tricks for improving the stability of training tree-based models.

SYMPOL is evaluated on several RL problems, largely simpler problems such as mountain car or lunar lander, as well as a MiniGrid environment. Finally, the interpretability of SYMPOL policies is displayed using a case study on policy generalization, in which the authors inspect a learned policy, identify areas for improvement or weaknesses in the policy, and then re-train to fix these weaknesses.

**Strengths:**

* The paper builds on very recent work and adequately cites the prior work, making sure to clearly draw the line between novel contributions and techniques which already exist.
* The SYMPOL framework is clearly explained, and the tricks used to improve training stability are highlighted (as opposed to being buried in the appendix). I feel like I could re-implement this from the paper alone.
* The results are compelling. SYMPOL performs quite well (final performance numbers/returns are superior to baselines) while also generalizing well to new versions of the problems (as shown by the Cohen's D experiments)
* The case study on interpretability is compelling, showing the power of an interpretable RL framework such as SYMPOL.

Overall, the paper is very well written, the framework is clearly described, makes intuitive sense, and does not require significant hand-tuning, and the results and discussion sections are strong.

**Weaknesses:**

* SYMPOL is mentioned to have 50+ nodes, which could become quite unwieldy or difficult to interpret, but we do not see such examples in the paper. It would be illustrative to see a standard SYMPOL policy in addition to the cleaned up versions in Figures 7/9.
* While the framework is sound and the results are strong, the overall novelty of the framework is limited. Apart from introducing a dynamically growing replay buffer, the technical contributions are largely a combination of existing techniques.
* As the paper relies heavily on the contribution of GradTree, it would be helpful to have an overview of that work here, so that the finished paper reads as a more self-contained work.
* The comparisons across all works are not entirely 1-1, as the number of nodes in each tree is not consistent (however, the authors make note of this in the text).

**Questions:**

* There is no ablation on the automated pruning, but automated pruning is mentioned a few times in the work. What are the effects of this on the resulting policy, and at what point are the node automatically pruned? If they are not pruned during training, do they complicate gradient flow?
* How is misaligned behavior detected, as in the case study shown in Section 6? It seems that, because the network's parameters are 1D feature vectors, you would need to reverse engineer the input states for each node, then plot out the resulting tree? Is there another way, or is this manually done?
* Would SYMPOL extend naturally to continuous action spaces or unstructured input (e.g., images)?

---

> ### Author Response · Authors · 2024-11-20
> **Rebuttal**
>
> We thank the reviewer for their time and valuable feedback!
>
> > SYMPOL is mentioned to have 50+ nodes, which could become quite unwieldy or difficult to interpret, but we do not see such examples in the paper. It would be illustrative to see a standard SYMPOL policy in addition to the cleaned up versions in Figures 7/9.
>
> We understand this concern. Unfortunately, it is difficult to visualize such policies in the context of a paper which is why we decided to use a comparatively small example. Nonetheless, we added an example of a large tree with 59 nodes for CartPole in the appendix (now Figure 10). While the tree is comparatively large, we can observe that the main logic is contained in the nodes at higher levels, focusing on the pole angle and the pole angular velocity. The less important features are in the lower levels where splits are often made on the cart position, which is not required to solve the task perfectly.  Frequently, additional nodes cover some minor additional differences (like first splitting Pole Angle < 0.0079, and then splitting Pole Angle < 0.0081).  This also highlights the potential for advanced post-hoc pruning methods to increase interpretability and potentially even generalization, which is subject to future work. However, we want to note that trees of this size are not specific to SYMPOL. Related methods have similar, often even higher tree sizes, as shown in Table 10.
>
> > While the framework is sound and the results are strong, the overall novelty of the framework is limited. Apart from introducing a dynamically growing replay buffer, the technical contributions are largely a combination of existing techniques.
>
> We agree that a core aspect of our method is the integration of GradTree into existing RL frameworks. While these methods are individually well-established, we believe that integrating GradTree into a dynamic task is a significant contribution. This integration has not been achieved previously and could have a substantial impact on the community, as well as future research in the active field of tree-based RL. Furthermore, directly applying GradTree within RL frameworks leads to unstable and ineffective outcomes, as the inherent instability in training tree-based models poses distinct difficulties in this setting.
>
> To achieve stability and effectiveness in RL, we developed several essential adjustments. As demonstrated in our ablation study (Figure 5), these modifications are crucial for SYMPOL to maintain interpretability and achieve robust performance under end-to-end gradient-based optimization. Without these adjustments, both stability and effectiveness in RL applications would be significantly compromised.
>
> However, with these adaptations in place, SYMPOL is framework-agnostic, allowing for straightforward integration into diverse RL algorithms beyond PPO. As illustrated in Section A.1 of the appendix, SYMPOL integrates seamlessly with A2C, demonstrating superior performance compared to alternative methods. This adaptability highlights the broad applicability and robustness of our approach across RL frameworks. In the revised version of the paper, we state our distinct contributions more explicitly in the introduction (L70-L78).
>
> > As the paper relies heavily on the contribution of GradTree, it would be helpful to have an overview of that work here, so that the finished paper reads as a more self-contained work.
>
> We agree that it is important to provide an overview of GradTree in this paper. To provide a better understanding, we included an additional figure (now Figure 2), to clarify how GradTree works. However, due to space constraints, it is unfortunately not possible to include a more detailed description of GradTree here.

---

### Official Review · Reviewer_7Ps3 · 2024-10-27

**Soundness:** 4
**Presentation:** 3
**Contribution:** 3
**Rating:** 10
**Confidence:** 5

**Summary:**

Authors propose SYMPOL. SYMPOL alleviates previous challenges of using RL to learn decision tree policies by using a differentiable tree representation. They also increase training stability by collecting longer rollouts at the end of training and using large batch sizes. The paper is well written. However, with the current experiments, it is unclear to me if the good experimental results are due to the use of GradTree or to the stabilizing of the training or both. But in any case, the contribution is not clear.

**Strengths:**

The related work and preliminary sections are excellent. Misgeneralization experiment and information loss experiments are insightful. Topic of interpretability is crucial.

**Weaknesses:**

There are two main weaknesses to this paper.

1) Contributions. Is the contribution plugging GradTree into PPO ?

I do find the ablation study insightful on why the training process needs strong stabilyzing and digging this further could be a nice contribution. Similarly, it seems to me that Silva et al 2020 is very similar to GradTree in the sense that GradTree is a differentiable formulation of DTs and Silva et al is too. I would have liked a more in depth comparison of the two!

2) Experiments (just the section 5.2). Baselines seem underperforming.

From looking at Silva et al 2020 and VIPER 2018, I feel like SA-DT (d=5) and D-SDT should be able to solve CartPole contrary to what you show in your table 1. Also why isn't SA-DT called VIPER?

If I were you I would rewrite the section 5 centered around two questions: why use direct tree optimization rather than post hoc distillation of a neural net like VIPER? After showing that direct optim might be better than post hoc fitting of trees, you should ask which of the two sota direct optim is better and why: Silva et al or SYMPOL?

**Questions:**

Could you remove fig 2 that does not bring anything to the paper and replace it with appendix results (such as figure 10)? Even better, could you summarize the section 4.1 or GradTree in a schematic and replace the current figure 2 with it please? If you decide to put figure 10 in the main paper instead of the schematic at figure 2, please smooth your curves.

Could you please remove figure 3 and replace it with somehting more useful? I think one does not need a plot to understand your rollout schedule.

Could you write a dedicated parargaph that explains the difference between SYMPOL and Silva et. al. 2020 ? I think your work is actually very similar to their.?

Bonus, add CUSTARD (Topin et al 2021) as a baseline in your table 1.

---

> ### Author Response · Authors · 2024-11-20
> **Rebuttal (Part 1 / 4)**
>
> We sincerely appreciate the reviewer’s time and thoughtful feedback! We will address your concerns and questions in the following:
>
>
> > However, with the current experiments, it is unclear to me if the good experimental results are due to the use of GradTree or to the stabilizing of the training or both. But in any case, the contribution is not clear.
>
> The good results originate from both the use of GradTree in combination with a stabilization of the training. Training stabilization is crucial to make GradTree work in a dynamic environment like RL, as shown in the ablation study (Figure 5). However, only GradTree can utilize the benefits of stabilization. It cannot be used to stabilize the training of alternative tree-based models. For these models, poor performance does not originate from instability during the training (as shown by the good results for MLP and SDT policies), but rather from information loss that occurs during distillation or discretization. However, it is not possible to integrate the proposed stabilization procedure into processes like distillation. This is because the model already uses the complete data from independent trajectories to fit the interpretable policy. Accordingly, the proposed stabilization can only be applied to SYMPOL and not to alternative methods for learning DT policies post-hoc. We agree that this should be explained in more detail and therefore clarified the distinct contributions of our method (L70-L86).
>
> > Contributions. Is the contribution plugging GradTree into PPO ? I do find the ablation study insightful on why the training process needs strong stabilyzing and digging this further could be a nice contribution.
>
> **TL;DR: Our approach extends beyond simply incorporating GradTree into PPO by developing essential adjustments that enable a stable, effective, and interpretable model and facilitate seamless integration into existing RL frameworks.**
>
> While our approach builds on the concept of gradient-based, axis-aligned, hard decision trees (GradTree), we extend it considerably to address the unique challenges of RL. Directly applying GradTree within RL frameworks results in unstable and ineffective outcomes, as the inherent instability in training tree-based models poses distinct difficulties in this setting.
>
> To achieve stability and effectiveness in RL, we developed several essential adjustments. As demonstrated in our ablation study (Figure 5), these modifications are crucial for SYMPOL to retain interpretability while achieving robust performance under end-to-end gradient-based optimization. Without these adjustments, both stability and effectiveness in RL applications would be significantly compromised.
>
> However, with these adaptations in place, SYMPOL is framework-agnostic, allowing for straightforward integration into diverse RL algorithms beyond PPO. As illustrated in Section A.1 of the appendix, SYMPOL integrates seamlessly with A2C, demonstrating superior performance compared to alternative methods. This adaptability highlights the broad applicability and robustness of our approach across RL frameworks. To clarify the conributions of you paper, we refined the corresponding paragraph in the introduction (L70-L86).
>
> The stabilization of the training process through dynamic buffer and batch sizes is a noteworthy finding from our study. We agree with the reviewer and are planning to investigate this component and the potential impact on the broader field of RL in future work.

---

> > ### Author Response · Authors · 2024-11-20
> > **Rebuttal (Part 2 / 4)**
> >
> > > Similarly, it seems to me that Silva et al 2020 is very similar to GradTree in the sense that GradTree is a differentiable formulation of DTs and Silva et al is too. I would have liked a more in depth comparison of the two!
> > >
> > > Could you write a dedicated parargaph that explains the difference between SYMPOL and Silva et. al. 2020 ? I think your work is actually very similar to their.?
> >
> > We agree that the paper would benefit from a more thorough and direct comparison of SYMPOL and Silva et al. (2020) to make the differences clearer. Therefore, we added a paragraph in the related work section of our revised paper to clearly distinguish SYMPOL from methods using differentiable decision trees (L138-L143; L247-L252).
> >
> > In related work, differentiable decision trees, such as those proposed by Silva et al. (2020), are typically soft decision trees (SDTs). SDTs achieve differentiability by relaxing discrete decisions related to feature and path selection at internal nodes. This approach is fundamentally different from SYMPOL, which does **not** use differentiable decision trees. Instead, SYMPOL leverages GradTree to optimize standard, non-differentiable decision trees through gradient descent. As a result, SYMPOL retains univariate, hard decisions, which are more interpretable than the oblique splits involving multiple features in SDTs and differentiable decision trees.
> >
> > For Silva et al. (2020), interpretability is achieved post-hoc by discretizing their soft decision tree model, which results in information loss and therefore a comparatively low performance as shown in our evaluation. SYMPOL directly optimizes the interpretable policy and, therefore, does not suffer information loss resulting in better performance.

---

> > > ### Author Response · Authors · 2024-11-20
> > > **Rebuttal (Part 3 / 4)**
> > >
> > > > Experiments (just the section 5.2). Baselines seem underperforming.
> > > From looking at Silva et al 2020 and VIPER 2018, I feel like SA-DT (d=5) and D-SDT should be able to solve CartPole contrary to what you show in your table 1. Also why isn't SA-DT called VIPER?
> > >
> > > **TL;DR: We clarified the distinction between SA-DT and VIPER and showed in an additional experiment that they yield similar results.**
> > >
> > > We understand that a reliable evaluation is essential to substantiate the claims in our paper. SA-DT (Silva et al. 2020) can be considered as a version of DAGGER and is conceptually similar to VIPER (Q-DAGGER) which improves data collection by incorporating additional weighting. We clarified this in the method section of the revised version of our paper (L343-L346).
> > >
> > > To address your concern and strengthen the evaluation, we have included an additional experimental comparison with original VIPER in our analysis (Section A.3) as well as below. Our results remain consistent with our original claims, demonstrating that SYMPOL outperforms these alternative approaches. This is also in-line with the results reported by `[1]`, where the authors show that the sampling in VIPER does not yield a better performance compared to DAGGER/SA-DT for interpretable DTs.
> > >
> > >
> > > |                          | CP   | AB   | LL   | MC-C | PD-C |
> > > |--------------------------|-------|-------|-------|-------|-------|
> > > | **SYMPOL (ours)**        | **500** | **-80**  | **-57**  | **94**  | **-323** |
> > > | D-SDT                    | 128   | -205  | -221  | -10   | -1343 |
> > > | SA-DT (d=5)              | 446   | -97   | -197  | **97**  | -1251 |
> > > | SA-DT (d=8)              | 476   | **-75**  | -150  | **96**  | -854  |
> > > | VIPER (d=5)              | 457   | **-77**  | -200  | -     | -     |
> > > | VIPER (d=8)              | 480   | **-75**  | -169  | -     | -     |
> > >
> > > **Control Performance.** We report the average undiscounted cumulative test reward over 25 random trials. The best interpretable method, and methods not statistically different, are marked bold. Please note that VIPER cannot be applied to continuous environments
> > >
> > > | Method              | E-R   | DK    | LG-5  | LG-7  | DS    |
> > > |---------------------|-------|-------|-------|-------|-------|
> > > | **SYMPOL (ours)**   | **0.964** | **0.959** | **0.951** | **0.953** | 0.939 |
> > > | D-SDT               | 0.662 | 0.654 | 0.262 | 0.381 | 0.932 |
> > > | SA-DT (d=5)         | 0.583 | **0.958** | **0.951** | 0.458 | **0.952** |
> > > | SA-DT (d=8)         | 0.845 | **0.961** | **0.951** | 0.799 | **0.954** |
> > > | VIPER (d=5)         | 0.651 | **0.958** | **0.948** | 0.456 | **0.954** |
> > > | VIPER (d=8)         | 0.845 | **0.963** | **0.948** | 0.801 | **0.954** |
> > >
> > > **MiniGrid Performance.** We report the average undiscounted cumulative test reward over 25 random trials. The best interpretable method, and methods not statistically different, are marked bold.
> > >
> > > The results reported in the original VIPER paper stating to achieve a perfect reward for CartPole are on a different version of the environment (CartPole-v0) with only 200 compared to 500 time steps and less randomness (CartPole-v1), making the underlying task easier. Also, we want to note that the reported results are in-line with related work, reporting comparable or worse results than those presented here. For instance, `[2]` report a mean reward of only 367 for VIPER on CartPole-v1. Also, `[3]` show poor performance of VIPER in general and specifically for LunarLander, the performance is worse than what we reported. Our findings align with these, suggesting that differences in performance may reflect randomness and lack of generalizability in the evaluation. We hope this clarifies our evaluation methodology and substantiates our comparative claims.

---

> > > > ### Author Response · Authors · 2024-11-20
> > > > **Rebuttal (Part 4 / 4)**
> > > >
> > > > > If I were you I would rewrite the section 5 centered around two questions: why use direct tree optimization rather than post hoc distillation of a neural net like VIPER? After showing that direct optim might be better than post hoc fitting of trees, you should ask which of the two sota direct optim is better and why: Silva et al or SYMPOL?
> > > >
> > > > We agree that addressing these two questions in our evaluation is essential and to clearly answer them restructured our evaluation accordingly. In general, post-hoc distillation comes with the problem of information loss, as shown in Table 3 and discussed in detail in Section A.2. This information loss often comes with a poor performance of post-hoc methods, as they are not able to capture the complexity of the originally trained model while still maintaining interpretability.
> > > >
> > > > Therefore, we believe that directly optimizing an intrinsically interpretable model is crucial. In the regime of direct optimization, SYMPOL is the first method that directly optimizes hard, axis-aligned and therefore easily interpretable models. Alternative methods, including Silva et al. (2020) optimize soft decision trees. In contrast to the hard decision trees optimized by SYMPOL, they often lack interpretability, as multiple features are considered at each split and additionally there are no hard decisions on which path to take. Therefore, to achieve a good interpretability, post-hoc distillation or discretization is also required for Silva et al., leading to the same issues discussed in the information loss section. To clarify this distinction, we included an additional paragraph in the related work section of our revised paper (L138-L143; L247-L252).
> > > >
> > > > > Could you remove fig 2 that does not bring anything to the paper and replace it with appendix results (such as figure 10)? Even better, could you summarize the section 4.1 or GradTree in a schematic and replace the current figure 2 with it please? If you decide to put figure 10 in the main paper instead of the schematic at figure 2, please smooth your curves.
> > > >
> > > > Based on the reviewers' feedback, we removed Figure 2 and included a modified version of originally Figure 10 from the appendix to the evaluation section (now Figure 3). We also smoothed the curves in the new Figure 3 as well as in  Figure 1 in the introduction. Furthermore, we included an additional figure explaining the representation of GradTree for a better understanding of the method, as also requested by other reviewers (now Figure 2).
> > > >
> > > > > Could you please remove figure 3 and replace it with somehting more useful? I think one does not need a plot to understand your rollout schedule.
> > > >
> > > > We agree that this figure is not necessary to understand the schedule and therefore removed it. The freed up space was used to thoroughly address the reviewers' concerns, e.g. for a more detailed comparison with Silva et al. (2020) in a separate paragraph (L138-L143) and moving the learning curves to the main paper (now Figure 3).
> > > >
> > > > > Bonus, add CUSTARD (Topin et al 2021) as a baseline in your table 1.
> > > >
> > > > We would be really eager to compare SYMPOL with CUSTARD, as it is highly related. Unfortunately, to the best of our knowledge, there is no existing official or proven implementation of CUSTARD that would facilitate a good comparison making a thorough comparison infeasible.
> > > >
> > > > `[1]` Kohler, Hector, et al. "Interpretable and Editable Programmatic Tree Policies for Reinforcement Learning." Workshop on Interpretable Policies in Reinforcement Learning@ RLC-2024.
> > > >
> > > > `[2]` Vos, Daniël, and Sicco Verwer. "Optimizing Interpretable Decision Tree Policies for Reinforcement Learning." arXiv, 21 Aug. 2024, https://arxiv.org/abs/2408.11632.
> > > >
> > > > `[3]` Kenny, Eoin M., et al. "Towards Interpretable Deep Reinforcement Learning with Human-Friendly Prototypes." International Conference on Learning Representations, 2023, https://openreview.net/pdf?id=hWwY_Jq0xsN.

---

> > > > > ### Comment · Reviewer_7Ps3 · 2024-11-21
> > > > > **Good work overall**
> > > > >
> > > > > I think your paper is good and I raised my score to 6. I am willing to raise it to 8 if you do the following:
> > > > >
> > > > >
> > > > > - You really have to center your contribution around figures 1 and 3 (in my humble opinion). So please make your figures more beautiful. I did not find an other wat to formulate this but for example figure 1 does not need thos big red arrows and boxes. Figure 3 legend are 'LL' why not write LunarLander come on ?
> > > > > - Assuming SDT/DDTs and GradTree are all just algorithms of the same class, i.e algorithms that allow direct optimization of trees, make it clear that SDT/DDTs are less interpretable and/or are harder to stabilize during training by also training some sdts with your stability tricks maybe reproducing your ablation fig 5 for sdts.
> > > > > - Make SYMPOL work on LunarLander.

---

> > > > > > ### Comment · Reviewer_7Ps3 · 2024-11-21
> > > > > > **Better plots**
> > > > > >
> > > > > > I recommend you to include the minigrid training curves test curves as well in the fashion of your figure 3.
> > > > > > You can use some standard rl library like https://github.com/google-research/rliable to plot performances of SYMPOL SDTs and MLPs.

---

> > > > > > ### Comment · Reviewer_7Ps3 · 2024-11-21
> > > > > > **Interpretability**
> > > > > >
> > > > > > As mentioned by other reviewers and yourself, SYMPOL trees are more interpretable than SDT for example but can also grow quite big for simple problems like cartpole. Can you add a paragrahp or an expirment ot convince us of the interpretability advantages of SYMPOL please ?

---

> > > > > > ### Comment · Reviewer_7Ps3 · 2024-11-21
> > > > > > **Opinion on contribution and presentation**
> > > > > >
> > > > > > In my honest opinion I think you should rewrite the paper to match the following presentation, of course this is up to you but I feel like as an expert in the field I would be more eager to read your work with the following high level changes:
> > > > > >
> > > > > > - Title: I am not sure symbolic has anything to do here. In fact in general I feel like titles starting by an algorithm name sound a bit cheap: you should title the scientific challenge you are tackling. Here I would say the challenge you are tackling is 'Understanding Direct Optimization of Tree-Based Reinforcement Learning: Generalization, Stability and Interpretability'
> > > > > >
> > > > > > - Don't center your contribution around SYMPOL which is gradtree + ppo + stabilizing tricks; center your contribution around the problem you are solving which is mitigating information loss of post-hoc methods with direct optim, but direct optim is hard to stabilize and non-interpretable for sdts, etc ...
> > > > > >
> > > > > > - Really make better plots

---

> > > > ### Comment · Reviewer_7Ps3 · 2024-11-21
> > > > **Thank you**
> > > >
> > > > Thank you for answering thoroughly to my review. I do feel like your work could benefit from better performances. I saw that some reviewers suggested to benchmark SYMPOL on OCAtari which is time consumming. Let us just say that if you manage to make SYMPOL work on LunarLander (get 200) OR explain why it fails it will really strengthen your paper.

---

> > > ### Comment · Reviewer_7Ps3 · 2024-11-21
> > > **Still not convinced**
> > >
> > > Well you do also have this dense tree architecture which is in my opinion another perspective on Silva's work and is not fundamentally different as you claim. But I might be wrong! Please convince me!
> > >
> > > I mean there is no need to think to hard to see that SYMPOL cannot actually solely retains a hard tree right? You do have to approximate or soften or densify the tree at some point.
> > >
> > > But it is not necessarily a bad thing. As I said in my original review your results still show that direct optimization (SYMPOL, Silva, Topin,...) are better than Dagger/VIPER.
> > >
> > > I forget if you mentioned this already but did you try SDTs/DDT with your stabilizing tricks?

---

> > ### Comment · Reviewer_7Ps3 · 2024-11-21
> > **Thank you**
> >
> > It is clear that stabilizing the training of the tree policy is challenging.

---

> ### Author Response · Authors · 2024-11-22
> **Extended Rebuttal (1/3)**
>
> >I think your paper is good and I raised my score to 6. I am willing to raise it to 8 if you do the following:
>
> Thank you for increasing your score and giving us the opportunity to refine our work further by incorporating your valuable feedback. Please find our responses below. Several adjustments have already been implemented in the revised PDF, with additional changes highlighted in red in the appendix. However, some changes require additional time, and we will include them as soon as possible.
>
>
> > Well you do also have this dense tree architecture which is in my opinion another perspective on Silva's work and is not fundamentally different as you claim. But I might be wrong! Please convince me!
>
> > I mean there is no need to think to hard to see that SYMPOL cannot actually solely retains a hard tree right? You do have to approximate or soften or densify the tree at some point.
>
> We acknowledge that the dense formulation shares similarities with Silva's work and other methods utilizing soft decision trees (SDTs). However, the key distinction in our dense representation lies in the use of two separate weight vectors at each internal node, rather than a single weight vector and bias. These two vectors correspond to the components of a standard axis-aligned split: (1) the threshold determining the decision boundary, and (2) the feature selected for splitting.
>
> The primary difference, however, stems from the training procedure, where our approach fundamentally diverges from Silva's work and SDTs/DDTs. Specifically, our method strictly retains a hard tree structure throughout training. Unlike existing methods that soften or approximate the tree during training, the hard tree structure in our model is preserved even during training. Softening is only introduced for gradient computation. When calculating the tree’s output, the structure remains hard and axis-aligned. For example, in Equation 5, we use the rounding operation to enforce hard decisions, which is excluded only during the backward pass where gradients are computed using the straight-through (ST) operator.
>
> Similarly, we apply the hardmax function to ensure that the split index vector remains one-hot encoded during the forward pass, maintaining univariate, axis-aligned splits. During the backward pass, this non-differentiable operation is excluded, allowing gradients to be computed for a soft model. However, in the forward pass, our model always retains a hard and axis-aligned structure due to the ST operator. This approach contrasts with existing methods, where the tree remains soft during both the backward and forward passes. Consequently, SYMPOL optimizes the exact hard and axis-aligned policy that will be applied later, eliminating the need for post-processing.

---

> > ### Author Response · Authors · 2024-11-22
> > **Extended Rebuttal (2/3)**
> >
> > >You really have to center your contribution around figures 1 and 3 (in my humble opinion). So please make your figures more beautiful. I did not find an other wat to formulate this but for example figure 1 does not need thos big red arrows and boxes. Figure 3 legend are 'LL' why not write LunarLander come on ?
> >
> > Thank you for your honest feedback. Our initial goal was to highlight the information loss more clearly in the images, but we understand that this made the figures less visually appealing. We have revised the figures accordingly the figures and hope they are now improved.
> >
> > >Assuming SDT/DDTs and GradTree are all just algorithms of the same class, i.e algorithms that allow direct optimization of trees, make it clear that SDT/DDTs are less interpretable and/or are harder to stabilize during training by also training some sdts with your stability tricks maybe reproducing your ablation fig 5 for sdts.
> > > As mentioned by other reviewers and yourself, SYMPOL trees are more interpretable than SDT for example but can also grow quite big for simple problems like cartpole. Can you add a paragrahp or an expirment ot convince us of the interpretability advantages of SYMPOL please ?
> >
> > Thank you for highlighting this. We initially included a visualization of an SDT for comparison, but removed it as it was overly complex for effective interpretation. However, in this context, we agree that providing a more detailed comparison would be valuable. Therefore, in addition to the two figures of SYMPOL already included in the paper, we have added a visualization of the corresponding SDT (Figure 11 and Figure 12) in the appendix (now Section A.5).
> >
> > The interpretability of SYMPOL and axis-aligned decision trees (DTs) is significantly greater than that of soft decision trees (SDTs), both at the level of individual splits and the overall tree structure. For example, when comparing the root nodes in the CartPole environment, the standard decision tree (Figure 10) uses a straightforward threshold: "𝑠_3 (Pole Velocity) ≤ -0.800". In contrast, the corresponding SDT (Figure 11) defines a much more complex decision boundary: "σ(−1.14𝑠_0 − 0.30𝑠_1 + 0.94𝑠_2 + 0.11𝑠_3 + 0.99)", which is far less intuitive.
> >
> > At a higher level, this difference in interpretability becomes even more pronounced when considering complete decision paths or the entire tree structure. SYMPOL employs deterministic splits, resulting in hard yes/no decisions at each node. In contrast, SDTs use probabilistic routing, introducing two additional disadvantages. First, the need to consider multiple paths simultaneously complicates the interpretation of the model's behavior. Second, the outputs of leaf nodes cannot be directly interpreted, as they are weighted by the probabilities of the paths leading to them. This comparison is further discussed in a brief paragraph in the appendix (now Section A.5).

---

> > > ### Author Response · Authors · 2024-11-22
> > > **Extended Rebuttal (3/3)**
> > >
> > > > I forget if you mentioned this already but did you try SDTs/DDT with your stabilizing tricks?
> > >
> > > We agree that exploring the stability procedure for alternative methods is an interesting topic. In general, MLPs and SDTs do not exhibit the same stability issues as SYMPOL without our adjustments. The increased instability in SYMPOL arises from discrete decisions made through hard, axis-aligned splits during training—a characteristic unique to this method. Nonetheless, we will re-run an ablation study on SDTs and MLPs using our stabilization procedure and will include a corresponding visualization and discussion in the appendix once the results are available.
> > >
> > > Additionally, we have included a detailed comparison of interpretability in the appendix (please see our response to your next question for further details).
> > >
> > >
> > >
> > >
> > > >I recommend you to include the minigrid training curves test curves as well in the fashion of your figure 3. You can use some standard rl library like https://github.com/google-research/rliable to plot performances of SYMPOL SDTs and MLPs.
> > >
> > > Thank you for your suggestion. We will include the training curves in the appendix of a revised version as soon as possible. For the layout of the plots, we have based our design on the shared library. The existing figures in the main paper and appendix have already been adjusted.
> > >
> > > We also appreciate you sharing the library for performance comparison. We plan to use it in future work and will attempt to incorporate it into the current paper during the rebuttal phase if feasible.
> > >
> > > > Thank you for answering thoroughly to my review. I do feel like your work could benefit from better performances. I saw that some reviewers suggested to benchmark SYMPOL on OCAtari which is time consumming. Let us just say that if you manage to make SYMPOL work on LunarLander (get 200) OR explain why it fails it will really strengthen your paper.
> > >
> > > In our paper, we argue that SYMPOL and decision tree-based policies, in general, are not well-suited for modeling control environments, such as those involving physical relationships. For simple tasks, this limitation is less critical. For instance, in environments like CartPole or MountainCar, relatively simple decision trees can solve the problem perfectly, although these remain challenging to learn. However, the situation is different for LunarLander. LunarLander is not typically classified as a control environment but instead shares similarities with object-centric Atari environments. For example, the lander's position is defined by its *x* and *y* coordinates, and its linear velocity is defined along the *x*- and *y*-axes. Such observations are not well-handled by hard, axis-aligned decision trees. Instead, oblique decisions are required in these environments, aligning with the argument presented by Kohler et al. (2024). This is further supported by results in related work, such as those reported by Silva et al. (2020), where hard, axis-aligned decision trees fail to achieve even a single positive reward (the best reported reward being -78 compared to -57 for SYMPOL).
> > >
> > >
> > >
> > > > In my honest opinion I think you should rewrite the paper to match the following presentation, of course this is up to you but I feel like as an expert in the field I would be more eager to read your work with the following high level changes:
> > >
> > > > Title: I am not sure symbolic has anything to do here. In fact in general I feel like titles starting by an algorithm name sound a bit cheap: you should title the scientific challenge you are tackling. Here I would say the challenge you are tackling is 'Understanding Direct Optimization of Tree-Based Reinforcement Learning: Generalization, Stability and Interpretability'
> > >
> > > Thank you for your suggestion. Unfortunately, it is not possible to change the title of the submission during the rebuttal phase. Title changes are only permitted for the camera-ready version. Based on your suggestion, we plan to revise the title for the camera-ready version to better reflect the problem we are addressing: "Mitigating Information Loss in Tree-Based Reinforcement Learning via Direct Optimization."
> > >
> > >
> > > > Don't center your contribution around SYMPOL which is gradtree + ppo + stabilizing tricks; center your contribution around the problem you are solving which is mitigating information loss of post-hoc methods with direct optim, but direct optim is hard to stabilize and non-interpretable for sdts, etc ...
> > >
> > > We agree with your suggestion and modified our contributions accordingly.

---

> ### Comment · Reviewer_7Ps3 · 2024-11-22
> **Very good !**
>
> Thank you for working so hard. I raised my score to 8.
>
> I will ask for this work to be highlighted (raise my score to 10) if:
> - Make SYMPOL work on control tasks be feeding linear combination (affine) of features to your model like Kohler and Delfosse 2024
> - Make a clean Python library open source for everyone to use SYMPOL; it should be compatible with gymnsaium

---

> > ### Author Response · Authors · 2024-11-26
> > **SYMPOL + (Affine) Linear Features**
> >
> > > Thank you for working so hard. I raised my score to 8.
> >
> > We sincerely thank the reviewer for their insightful discussion, constructive feedback, and for acknowledging the improvements to the paper by raising the score.
> >
> > > I will ask for this work to be highlighted (raise my score to 10) if:
> >
> > > Make SYMPOL work on control tasks be feeding linear combination (affine) of features to your model like Kohler and Delfosse 2024
> >
> > We appreciate the suggestion regarding the inclusion of (affine) linear combinations of features as an extension to SYMPOL. This modification addresses the challenges SYMPOL faces in environments such as LunarLander. Following your advice, we integrated (affine) linear combinations of features into our model, inspired by the approach of Kohler and Delfosse (2024). This adjustment significantly improved SYMPOL's performance in control environments. For example, on the *LunarLander* task, the average reward increased substantially from -57 to 105.
> >
> > Notably, the agent is now capable of solving the environment (achieving a reward > 200) in 10/25 test episodes, compared to only 1/25 test episode without the inclusion of affine features. Importantly, this improvement was achieved without performing new HPO. We believe the primary limitation currently lies in the stability of training and, at times, suboptimal generalization to test environments. Conducting new HPO tailored specifically for models incorporating (affine) linear features is likely to resolve these remaining issues.
> >
> >
> > |          | LunarLander   |
> > | -------- | -------- |
> > | SYMPOL     | -57     |
> > | SYMPOL + (Affine) Linear Features     | **105**     |
> >
> >
> > We plan to finalize our experiments, including HPO with (affine) linear combinations of features, across all control environments before submitting the camera-ready version. The results will be summarized in an additional section in the appendix.
> >
> > Furthermore, we are actively working on applying SYMPOL with (affine) linear features to OCAtari environments, and preliminary results are promising.
> >
> >
> > >Make a clean Python library open source for everyone to use SYMPOL; it should be compatible with gymnsaium
> >
> > We appreciate the recognition of this as a valuable contribution to the field. In line with this, we have already begun developing an open-source library that can be seamlessly integrated with gymnasium. Currently, our method is implemented within CleanRL, as provided in the supplementary material. This setup allows for easy reproduction of our results and is compatible with both gym/gymnasium and gymnax environments.
> >
> > Given the modular design of the current implementation, we believe it provides a solid foundation for building a clean and user-friendly library. Upon acceptance of the paper, we will make the library publicly available and easily accessible through PyPi. Additionally, we are working on integrating the library with stable_baselines3 to ensure the method is accessible and straightforward to use, even for non-experts.

---

> > > ### Comment · Reviewer_7Ps3 · 2024-11-27
> > > **Amazing**
> > >
> > > Don't forget to update the rebuttal summary for the AC.

---

### Official Review · Reviewer_6L9X · 2024-10-29

**Soundness:** 3
**Presentation:** 2
**Contribution:** 3
**Rating:** 8
**Confidence:** 4

**Summary:**

This work presents SYMPOL, a method for online RL using decision trees as policies.
The authors include the GradTree into PPO, thus backpropagating from a neural critic to an arithmetic decision tree-based actor.
They evaluate their method on different continuous and discrete environments, and demonstrate that SYMPOL allow to detect and correct misalignments problem that SIMPOL allows to find.

**Strengths:**

**The method is well motivated.**
GBRL, the only interpretable tree-based method that directly learns tree instead of distilling them after training, leads to huge and many trees.

**The potential impact of online learning of trees is huge for interpretable RL.**
The online learning of transparent policies could allow for detection of misalignment during training, and potentially correction of the misalignment during training. Distillation methods do not have this advantage.

**The structure of the paper makes it easy to follow**. Most of the paper is clear, in terms of structure, particularly the experimental evaluation. The bold points, together with the clear figures and tables, allow the reader to quickly grasp most of the experimental evaluation, even if it took me a bit of time to understand that SA-DT is an implementation of Dagger/VIPER.

**Weaknesses:**

**The difference between SYMPOL and other baselines is not clear.**
The preliminaries section would benefit for a bigger explanation on GradTree, as its integration to PPO is the main contribution and what distinguish this work from many existing interepretable baselines such as NUDGE, INSIGHT, GBRL and SCoBots ([1]).
It also took me quite some time to understand that SA-DT is equivalent to running Dagger or VIPER with their MLP.

**Evaluation seems weird.** It feels like the baselines are underperforming, compared to what is reported in the literature. For example, the D-SDT is underperforming compared to what VIPER reports (with a few node). I don't know why, as it is okay if their baselines would perform worse than distilled ones. Directly learning trees in an online fashion, could allow for e.g. LLM guidance to provide the trees with correct inductive biases, that would avoid e.g. misalignment problems during learning.

**Missing litterature on misgeneralisation (and interpretable RL).**
On misgeneralisation, NNs have been shown to systematically misgeneralize on different Atari environment [2], even when the environment is simplified [3], and DT-based policies have been shown to allow to mitigate it [1].


[1] Delfosse et al. "Interpretable concept bottlenecks to align reinforcement learning agents." Advances in Neural Information Processing Systems 37 (2024).

[2] Farebrother et al. Generalization and regularization in DQN. arXiv (2018).

[3] Delfosse et al. "HackAtari: Atari Learning Environments for Robust and Continual Reinforcement Learning." arXiv (2024).

**Questions:**

* Why is it that the extracted trees are performing so low in comparison to what is reported by the VIPER's authors ?
* Is there any reason why you are not using the set of Object Centric Atari environments in your evaluation ? It would allow to compare with other deep and interpretable baselines such as NUDGE, INSIGHT, SCoBots, ...

I'm willing to revise my scores if experiments that compare SYMPOL to these interpretable methods on some Atari games are reported. It would allow the reader to compare these methods. Even if SYMPOL slightly underperforms, for interpretability, game scores are not the most important focus.

---

> ### Author Response · Authors · 2024-11-20
> **Rebuttal (Part 1 / 3)**
>
> We sincerely thank the reviewer for their time and insightful feedback!
>
> > The difference between SYMPOL and other baselines is not clear. The preliminaries section would benefit for a bigger explanation on GradTree, as its integration to PPO is the main contribution and what distinguish this work from many existing interepretable baselines such as NUDGE, INSIGHT, GBRL and SCoBots ([1]).
>
> The main difference between SYMPOL and the mentioned baselines `[1,4,5,6]` is the conceptual integration into existing RL frameworks. While existing methods that integrate symbolic policies into existing frameworks require modifications of the RL framework itself, SYMPOL does not require such modifications. By utilizing GradTree, a DT policy can be learned directly from the policy gradients, making the proposed method framework-agnostic.
>
> We clarified the distinction of SYMPOL from alternative methods at the corresponding positions in the revised version of the paper (L180-L184). Additionally, we extended the explanation of GradTree including additional figures to give a better understanding of the method (L213-L227; L247-L252).
>
>
> > It also took me quite some time to understand that SA-DT is equivalent to running Dagger or VIPER with their MLP.
>
> In the revised version, we clarified the relationship between SA-DT and VIPER, which was not explicitly outlined in the initial submission (L343-L346). In general, SA-DT can be considered as a version of DAGGER. Therefore, SA-DT and VIPER are conceptually similar, but VIPER (Q-DAGGER) includes advanced sampling and weighting strategies for training samples — techniques not incorporated in SA-DT, as used, for example, by Silva et al. (2020). Furthermore, we included an additional experiment directly comparing VIPER, SA-DT and SYMPOL (Section A.3). The conclusions remain consistent with the ones presented in the initial version of the paper.

---

> > ### Author Response · Authors · 2024-11-20
> > **Rebuttal (Part 2 / 3)**
> >
> > > Evaluation seems weird. It feels like the baselines are underperforming, compared to what is reported in the literature. For example, the D-SDT is underperforming compared to what VIPER reports (with a few node). I don't know why, as it is okay if their baselines would perform worse than distilled ones.
> > > Why is it that the extracted trees are performing so low in comparison to what is reported by the VIPER's authors ?
> >
> > **TL;DR: We clarified the distinction between SA-DT and VIPER and showed in an additional experiment that they yield similar results.**
> >
> > We understand that a reliable evaluation is essential to substantiate the claims in our paper. To address your concern and strengthen the evaluation, we have included an additional experimental comparison with original VIPER in our analysis (Section A.3) as well as below. Our results remain consistent with our original claims, demonstrating that SYMPOL outperforms these alternative approaches. This is also in-line with the results reported by `[7]`, where the authors show that the sampling in VIPER does not yield a better performance compared to DAGGER/SA-DT for interpretable DTs.
> >
> > |                          | CP   | AB   | LL   | MC-C | PD-C |
> > |--------------------------|-------|-------|-------|-------|-------|
> > | **SYMPOL (ours)**        | **500** | **-80**  | **-57**  | **94**  | **-323** |
> > | D-SDT                    | 128   | -205  | -221  | -10   | -1343 |
> > | SA-DT (d=5)              | 446   | -97   | -197  | **97**  | -1251 |
> > | SA-DT (d=8)              | 476   | **-75**  | -150  | **96**  | -854  |
> > | VIPER (d=5)              | 457   | **-77**  | -200  | -     | -     |
> > | VIPER (d=8)              | 480   | **-75**  | -169  | -     | -     |
> >
> > **Control Performance.** We report the average undiscounted cumulative test reward over 25 random trials. The best interpretable method, and methods not statistically different, are marked bold. Please note that VIPER cannot be applied to continuous environments
> >
> > | Method              | E-R   | DK    | LG-5  | LG-7  | DS    |
> > |---------------------|-------|-------|-------|-------|-------|
> > | **SYMPOL (ours)**   | **0.964** | **0.959** | **0.951** | **0.953** | 0.939 |
> > | D-SDT               | 0.662 | 0.654 | 0.262 | 0.381 | 0.932 |
> > | SA-DT (d=5)         | 0.583 | **0.958** | **0.951** | 0.458 | **0.952** |
> > | SA-DT (d=8)         | 0.845 | **0.961** | **0.951** | 0.799 | **0.954** |
> > | VIPER (d=5)         | 0.651 | **0.958** | **0.948** | 0.456 | **0.954** |
> > | VIPER (d=8)         | 0.845 | **0.963** | **0.948** | 0.801 | **0.954** |
> >
> > **MiniGrid Performance.** We report the average undiscounted cumulative test reward over 25 random trials. The best interpretable method, and methods not statistically different, are marked bold.
> >
> > The results reported in the original VIPER paper stating to achieve a perfect reward for CartPole are on a different version of the environment (CartPole-v0) with only 200 compared to 500 time steps and less randomness (CartPole-v1), making the underlying task easier. Also, we want to note that the reported results are in-line with related work, reporting comparable or worse results than ours. For instance, `[8]` report a mean reward of only 367 for VIPER on CartPole-v1. Also, `[9]` show poor performance of VIPER in general and specifically for LunarLander the performance is worse than what we reported. Our findings align with these, suggesting that differences in performance may reflect randomness and lack of generalizability in the evaluation. We hope this clarifies our evaluation methodology and substantiates our comparative claims.

---

> > > ### Author Response · Authors · 2024-11-20
> > > **Rebuttal (Part 3 / 3)**
> > >
> > > > Directly learning trees in an online fashion, could allow for e.g. LLM guidance to provide the trees with correct inductive biases, that would avoid e.g. misalignment problems during learning.
> > >
> > > Including LLM guidance during training to provide correct inductive biases is an interesting idea, which we will take a closer look at in future work on real-world tasks.
> > >
> > > > Missing litterature on misgeneralisation (and interpretable RL). On misgeneralisation, NNs have been shown to systematically misgeneralize on different Atari environment [2], even when the environment is simplified [3], and DT-based policies have been shown to allow to mitigate it [1].
> > >
> > > Thank you very much for highlighting these relevant related works. While comparing our method with those works and evaluating misgeneralization in their setup is out of scope for this paper, we agree that it is crucial to discuss this and therefore have included the references and a short additional discussion in the revised version of our paper (L474-L475; L518-L520).
> > >
> > > > Is there any reason why you are not using the set of Object Centric Atari environments in your evaluation ? It would allow to compare with other deep and interpretable baselines such as NUDGE, INSIGHT, SCoBots, ...
> > >
> > > Thank you for this insightful question. We opted not to include the (Object-Centric) Atari environments because prior research `[6]` has already demonstrated that end-to-end trained DTs are not well-suited for continuous environments like Atari games. These environments, while valuable for deep learning approaches, are less effective for interpretable decision tree methods, which excel in different types of tasks like categorical environments.  Similarly, `[7]` states that using oblique decisions is essential to achieve reasonable performance on Object Centric Atari which precludes an efficient usage of SYMPOL, as the learned decision trees are axis-aligned. Consequently, we focused on alternative environments (like categorical MiniGrid environments) where DT-based methods show stronger performance, aiming to highlight their unique interpretability advantages.
> > >
> > > This, however, does not exclude real-world applications of the proposed method. Suitable scenarios could be traffic-light control `[10,11]` or healthcare `[12,13]` and medical diagnosis `[14,15]`, where interpretability offers practical insights and transparency. Evaluating such use cases and quantifying the real-world impact of enhanced interpretability is an avenue for future research.
> > >
> > > `[1]` Delfosse et al. "Interpretable concept bottlenecks to align reinforcement learning agents." Advances in Neural Information Processing Systems 37 (2024).
> > >
> > > `[2]` Farebrother et al. Generalization and regularization in DQN. arXiv (2018).
> > >
> > > `[3]` Delfosse et al. "HackAtari: Atari Learning Environments for Robust and Continual Reinforcement Learning." arXiv (2024).
> > >
> > > `[4]` Delfosse, Quentin, et al. "Interpretable and Explainable Logical Policies via Neurally Guided Symbolic Abstraction." arXiv, 25 Oct. 2023, https://arxiv.org/abs/2306.01439.
> > >
> > > `[5]` Luo, Lirui, et al. "End-to-End Neuro-Symbolic Reinforcement Learning with Textual Explanations." arXiv, 13 June 2024, arxiv.org/abs/2403.12451.
> > >
> > > `[6]` Fuhrer, Benjamin, et al. "Gradient Boosting Reinforcement Learning." arXiv, 11 July 2024, https://arxiv.org/abs/2407.08250.
> > >
> > > `[7]` Kohler, Hector, et al. "Interpretable and Editable Programmatic Tree Policies for Reinforcement Learning." Workshop on Interpretable Policies in Reinforcement Learning@ RLC-2024.
> > >
> > > `[8]` Vos, Daniël, and Sicco Verwer. "Optimizing Interpretable Decision Tree Policies for Reinforcement Learning." arXiv, 21 Aug. 2024, https://arxiv.org/abs/2408.11632.
> > >
> > > `[9]` Kenny, Eoin M., et al. "Towards Interpretable Deep Reinforcement Learning with Human-Friendly Prototypes." International Conference on Learning Representations, 2023, https://openreview.net/pdf?id=hWwY_Jq0xsN.
> > >
> > > `[10]` Wei, Hua, et al. "Intellilight: A reinforcement learning approach for intelligent traffic light control." Proceedings of the 24th ACM SIGKDD international conference on knowledge discovery & data mining. 2018.
> > >
> > > `[11]` Yau, Kok-Lim Alvin, et al. "A survey on reinforcement learning models and algorithms for traffic signal control." ACM Computing Surveys (CSUR) 50.3 (2017): 1-38.
> > >
> > > `[12]` Yu, Chao, et al. "Reinforcement learning in healthcare: A survey." ACM Computing Surveys (CSUR) 55.1 (2021): 1-36.
> > >
> > > `[13]` Nambiar, Mila, et al. "Deep offline reinforcement learning for real-world treatment optimization applications." Proceedings of the 29th ACM SIGKDD Conference on Knowledge Discovery and Data Mining. 2023.
> > >
> > > `[14]` Yu, Zheng, et al. "Deep reinforcement learning for cost-effective medical diagnosis." arXiv preprint arXiv:2302.10261 (2023).
> > >
> > > `[15]` Fatemi, Mehdi, et al. "Medical dead-ends and learning to identify high-risk states and treatments." Advances in Neural Information Processing Systems 34 (2021): 4856-4870.

---

> ### Comment · Reviewer_6L9X · 2024-11-23
> **NUDGE, INSIGHT and SCoBots do not require adapting the RL framework**
>
> Thank you for your clarification. I checked the paper, the provided modifications improve it a lot, and you clarified the difference between SA-DT and VIPER.
>
> However, I think that the claim that *SYMPOL is the only RL framework-agnostic method* is wrong.
> * The differentiable logic policies of NUDGE are optimized using the policy gradients
> * INSIGHT and SCoBots simply extract EQL and Object-centric DT from policy gradients' optimized NNs
>
> This claim would anyway not change my ratings, as I consider SYMPOL a nice contribution to the Interpretable RL field.
>
> Finally, the OCAtari environments are not much more "continuous" than the minigrid ones. Take *Pong* for example, if you consider the 3 relevant objects: the player, enemy, and the ball. The x position of the enemy and the player being fixed, (they can be discarded, as explained by [7]).
>
> You thus end up with 4 variables: Player.y, Enemy.y, Ball.x, Ball.y. These values have a discrete range of ~100 values.
>
> Eventually you can add dx and dy for each object (with even smaller range).
>
> > Would SYMPOL not be applicable to such an environment ?
>
> I consider that **your method does not need extra evaluation to confirm the paper's claim**, and that the claims are enough to make it accepted, so don't feel pressure to run these experiments, I am just trying to understand if someone could run them if they wanted to compare SYMPOL to other interpretable RL method.
>
> Thanks for your answer, I am already raising my score.

---

> > ### Author Response · Authors · 2024-11-25
> > **Further Clarification**
> >
> > > Thank you for your clarification. I checked the paper, the provided modifications improve it a lot, and you clarified the difference between SA-DT and VIPER.
> >
> > Thank you for acknowledging the improvements and clarifications. We are glad that the modifications we made based on your feedback have further enhanced the quality of the paper, and we also believe it is now significantly improved.
> >
> > > However, I think that the claim that SYMPOL is the only RL framework-agnostic method is wrong.
> >
> > >The differentiable logic policies of NUDGE are optimized using the policy gradients
> > INSIGHT and SCoBots simply extract EQL and Object-centric DT from policy gradients' optimized NNs
> > This claim would anyway not change my ratings, as I consider SYMPOL a nice contribution to the Interpretable RL field.
> >
> > Thank you once again for highlighting this point. You are indeed correct — while SYMPOL is framework-agnostic, it is not the only framework-agnostic method in this context. Our intention was to emphasize that SYMPOL can be seamlessly integrated into existing RL frameworks without requiring any modifications to the method or the underlying RL algorithm.
> >
> > > Finally, the OCAtari environments are not much more "continuous" than the minigrid ones. Take Pong for example, if you consider the 3 relevant objects: the player, enemy, and the ball. The x position of the enemy and the player being fixed, (they can be discarded, as explained by [7]).
> >
> > > You thus end up with 4 variables: Player.y, Enemy.y, Ball.x, Ball.y. These values have a discrete range of ~100 values.
> >
> > > Would SYMPOL not be applicable to such an environment ?
> >
> > Thank you for your feedback regarding OCAtari. Considering the variables as features with discrete ranges, we agree that SYMPOL could indeed be applied to such an environment. However, we anticipate that the resulting decision trees might become comparatively large to adequately represent all relevant scenarios. In contrast, oblique decision trees, such as those produced by INTERPRETER, may yield much more compact trees by directly modeling relationships between variables (e.g., *Player.y* − *Ball.y*).
> >
> > Currently, SYMPOL does not efficiently model such relationships. Nonetheless, we believe that extending SYMPOL to capture additional relationships while preserving its interpretability could be a promising direction for future work.

---

### Official Review · Reviewer_sgid · 2024-11-09

**Soundness:** 2
**Presentation:** 2
**Contribution:** 2
**Rating:** 5
**Confidence:** 4

**Summary:**

This paper proposes a differentiable tree-structured model that can be optimized via RL algorithms. The proposed method includes a tree-structure actor and a neural network critic. Experiments are conducted on several RL benchmarks include discrete and continuous action spaces, compared with previous tree-based methods.

**Strengths:**

1. The proposed method can be applied into both discrete and continuous action spaces.

2. The proposed method improves interpretability of RL policies.

**Weaknesses:**

The reviewer is not convinced with the claim of "first method that trains a DT via stochastic policy gradient in an end-to-end manner". Making the learning of DTs differentiable has been investigated for a long time, such as [1]. It is safe to make this claim through a thorough literature review.

There are some other works focusing on learning a differentiable symbolic policy which is based on first-order logic, such as [2,3], which is not compared and discussed in this paper.

Looks like that defining the syntax and semantics of DTs relies on human knowledge, maybe the reviewer is wrong, which can be clarified during rebuttal.

[1] Treeqn and atreec: differentiable tree-structured models for deep reinforcement learning.

[2] Neural logic reinforcement learning. 2019.

[3] GALOIS: boosting deep reinforcement learning via generalizable logic synthesis. 2022.

**Questions:**

The reviewer is confused about the information loss experiments, described in Fig 1 and Table 3. Is the test reward on the same environment as the trained one? If so, why is there a performance drop of (Silva et al. (2020))? Did you mean the trained model has a stochastic probability while the test model is greedy that is, selecting the action with the maximum Q?

---

> ### Author Response · Authors · 2024-11-20
> **Rebuttal (Part 1 / 2)**
>
> We appreciate the reviewer’s time and thoughtful feedback!
>
> > The reviewer is not convinced with the claim of "first method that trains a DT via stochastic policy gradient in an end-to-end manner". Making the learning of DTs differentiable has been investigated for a long time, such as [1]. It is safe to make this claim through a thorough literature review.
>
> **TL;DR: SYMPOL does not learn differentiable or soft DTs, but instead maintains univariate, hard decisions during training and inference. Clarifying paragraph was added to the revised paper.**
>
> We understand the concern raised by the reviewer and agree that this needs to be explained more thoroughly. Therefore, we have added an additional paragraph in the related work section distinguishing SYMPOL from existing methods involving Differentiable / Soft Decision Trees (L138-L143). We have also included the mentioned relevant references to the related work section (L131).
>
> In related work, including `[1]`, differentiable decision trees typically correspond to soft decision trees (SDTs), which achieve differentiability by relaxing discrete decisions related to feature and path selection at internal nodes. This approach is fundamentally different from SYMPOL, which does **not** use differentiable decision trees. Instead, SYMPOL leverages GradTree to optimize standard, non-differentiable decision trees through gradient descent. As a result, SYMPOL retains univariate, hard decisions, which are more interpretable than the oblique splits involving multiple features in SDTs and differentiable decision trees.
>
> However, we agree that our initial claim might be misleading. We do not assert that ours is the first work that trains a DT via stochastic policy gradient in an end-to-end manner. As the reviewer notes, this approach has been frequently explored in related work. However, existing work typically focuses on Differentiable or Soft DTs for end-to-end learning, whereas we utilize hard, axis-aligned DTs while still preserving gradient-based optimization. To clarify, we have adjusted the claim in the abstract of the revised paper (L20-L23) and added a more detailed discussion and comparison with existing work in the related work section (L138-L143; L247-L252).
>
> > There are some other works focusing on learning a differentiable symbolic policy which is based on first-order logic, such as [2,3], which is not compared and discussed in this paper.
>
> **TL;DR: We focus on DTs providing greater flexibility through learnable feature-threshold comparison. References and discussion have been added to the revised paper.**
>
> Thank you for highlighting this relevant work. We have added the suggested references and included a comparison in our related work section. Both NLRL [2] and GALIOS [3] aim to synthesize symbolic policies using logical rules, leveraging differentiable inductive logic programming. This allows for gradient-based optimization, similar to SYMPOL. The main distinction lies in the type of symbolic policy optimized by each approach: SYMPOL learns a decision tree, whereas NLRL and GALIOS learn sets of logical rules. Although both approaches share similarities, DTs offer greater flexibility by not only combining atomic conditions but also comparing features against thresholds — a critical capability for handling continuous observation spaces. As noted in the related work section, this paper focuses on learning DT-based policies, so a direct experimental comparison with rule-based methods is beyond the scope of our work. Nevertheless, we briefly discuss these methods in the revised related work section (L100-L105).
>
> > Looks like that defining the syntax and semantics of DTs relies on human knowledge, maybe the reviewer is wrong, which can be clarified during rebuttal.
>
> **TL;DR: SYMPOL does not require human knowledge for defining the syntax and semantics of the DT.**
>
> When learning a DT with SYMPOL, no human knowledge or human interaction is required. The user only has to define the maximum depth of the tree which can be seen as a hyperparameter (or simply fixed to a high value like 10). The entire DT as RL policy is then learned end-to-end with gradient descent and the learned policy (which is a decision tree as for instance in Figure 4) can be inspected by the user. We hope that our answer clarified this. If there are any concerns remaining, please let us know so that we can refine our paper to avoid future confusion.

---

> ### Author Response · Authors · 2024-11-20
> **Rebuttal (Part 2 / 2)**
>
> > The reviewer is confused about the information loss experiments, described in Fig 1 and Table 3. Is the test reward on the same environment as the trained one? If so, why is there a performance drop of (Silva et al. (2020))? Did you mean the trained model has a stochastic probability while the test model is greedy that is, selecting the action with the maximum Q?
>
> **TL;DR: Training evaluates original models (e.g., neural networks for SA-DT), while testing uses interpretable, deterministic models (e.g., distilled DTs) on distinct random seeds. Clarifications and additional training curves have been added to the revised paper.**
>
> Thank you for your feedback and for highlighting the need for clarification regarding the information loss experiments in Figure 1 and Table 3. To clarify, the test environments indeed differ from the training environments due to distinct random seeds used in the test phase. This explains why the rewards are not identical for SYMPOL. For other methods, however, the differences in performance additionally arise from the contrasting models used in training and testing phases. Specifically, during training, loss is reported on the model in its original form (e.g., the neural network policy for SA-DT), while during testing, we evaluate the interpretable model (e.g., the distilled decision tree for SA-DT), which operates in a deterministic manner.
>
> We acknowledge that these details were not sufficiently explained in the original figure and have revised the paper accordingly to make this distinction clearer (L35-L39). Also, we included an additional figure on the training curves in the main paper (now Figure 3) which should provide additional clarification.
>
> `[1]` Farquhar, Gregory, et al. "Treeqn and atreec: Differentiable tree-structured models for deep reinforcement learning." arXiv preprint arXiv:1710.11417 (2017).
>
> `[2]` Jiang, Zhengyao, and Shan Luo. "Neural logic reinforcement learning." International conference on machine learning. PMLR, 2019.
>
> `[3]` Cao, Yushi, et al. "GALOIS: boosting deep reinforcement learning via generalizable logic synthesis." Advances in Neural Information Processing Systems 35 (2022): 19930-19943.

---

> > ### Comment · Reviewer_sgid · 2024-11-24
> > **Thanks for the response**
> >
> > Thanks for the clarification. One more confusing part about the test results: the authors mentioned the test environments differ from the training ones due to the different random seeds, which is not the common setting from the reviewer's perspective. The random seed will only influence the trained neural network models, how does it affect the environment? If the difference only occurs in random seeds, does this mean the proposed method is more stable regarding different random seeds? But this stableness lies in deep learning architectures or some training tricks, right?

---

> > > ### Author Response · Authors · 2024-11-25
> > > **Further Clarification**
> > >
> > > We apologize for any confusion caused. We acknowledge that referencing different seeds in this context might be unclear. To clarify, we were *not* referring to the *model seeds*, but rather to the *environment seed* used during the initial reset of the environment. This seed controls the randomness within the environment, particularly the starting conditions. For instance, in the CartPole environment, it determines the initial pole position, while in MiniGrid, it may control the goal position.
> > >
> > > By selecting different *environment seeds*, we ensure that the reported performance of the agent does not stem from memorization of specific trajectories. Instead, this approach allows us to evaluate whether the model truly generalizes to the overall task and accurately measures its capability to solve the task under varying initial conditions. This refers to the *evaluation after learning* and should cover multiple runs and rollouts on test environments with different seeds `[1]`.
> > >
> > > It is important to note that this does *not* imply that the proposed method is inherently more robust to variations in model initialization. Rather, it demonstrates that our method generalizes effectively across different environment initializations.
> > >
> > > `[1]` Chan, Stephanie C.Y., et al. "Measuring the Reliability of Reinforcement Learning Algorithms." International Conference on Learning Representations, 2020, https://openreview.net/forum?id=SJlpYJBKvH.

---

> > > > ### Author Response · Authors · 2024-11-28
> > > > **Follow-Up on Review**
> > > >
> > > > As the review period is nearing its end, we wanted to check if you have any further questions or concerns about our submission. Additionally, we would like to ask if you might consider adjusting your score based on how we have addressed your feedback and the general improvement of our revised manuscript (see general response).
> > > >
> > > > Thank you for your time and efforts in reviewing our work!

---

### Official Review · Reviewer_DZ3Q · 2024-11-10

**Soundness:** 3
**Presentation:** 2
**Contribution:** 2
**Rating:** 5
**Confidence:** 5

**Summary:**

This work employs a tree-based model integrated with on-policy RL, like PPO, to maintain better interpretability with less information loss.
On a set of benchmark RL tasks, it demonstrates its superiority over alternative tree-based RL approaches in terms of performance and interoperability.

**Strengths:**

1. The motivation is good, aiming to improve the interpretability of RL, which is a key factor for deployment due to its adoption of DNNs.
2. The authors try to improve the stability of DTs as a policy via more implemented skills.

**Weaknesses:**

1. This work evaluates its experiments on a series of toy tasks that could be easily solved. Can it achieve competitive performance in complex tasks, like Mujoco and Atari games?
2. This work is also hard to follow, the contribution and the method should be highlighted and stated.
3. Please abatement the contribution of this work. This work is not the first to use DT to represent RL policy, i.e., the work "MIXRTs: Toward interpretable multi-agent reinforcement learning via mixing recurrent soft decision trees", arXiv:2305.10091, 2022. They utilized SDT in multi-agent RL complex tasks two years ago.
4. There are some typos in References.

**Questions:**

1. line 414- 415: "In RL, especially for smaller policies, the runtime is mainly determined by the actor-environment interaction..." - This sentence is hard to follow. Can the authors unpack this sentence?
2. Did this work simply incorporate GradTree into the PPO framework?

---

> ### Author Response · Authors · 2024-11-20
> **Rebuttal (Part 1 / 2)**
>
> We thank the reviewer for their time and valuable feedback!
>
> > This work evaluates its experiments on a series of toy tasks that could be easily solved. Can it achieve competitive performance in complex tasks, like Mujoco and Atari games?
>
> **TL;DR:SYMPOL excels in moderate-complexity tasks, where interpretability is crucial, and also holds potential for real-world applications in domains such as reinforcement learning for healthcare and traffic control.**
>
> While SYMPOL is technically adaptable to complex environments, we prioritize tasks where interpretability is both feasible and provides actionable insights. In high-complexity settings like Mujoco or Atari games, achieving holistic interpretability becomes challenging due to the model size and complexity required. The demonstrated strength of SYMPOL lies in tackling scenarios of moderate complexity, where its interpretability provides tangible benefits.
>
> This, however, does not exclude real-world applications of the proposed method. Suitable scenarios could be traffic-light control `[1,2]` or healthcare `[3,4]` and medical diagnosis `[5,6]`, where interpretability offers practical insights and transparency. Evaluating such use cases and quantifying the real-world impact of enhanced interpretability is an avenue for future research.
>
> Future research could explore an ensemble-based approach, such as Gradient-Boosted Reinforcement Learning (GBRL), to enhance the scalability of SYMPOL for more complex tasks. However, evidence from GBRL suggests that tree-based methods are generally not well-suited for environments with unstructured data domains, such as Atari games. Similarly, tasks requiring complex physical modeling, like Mujoco, are more effectively addressed by neural network policies.
>
> > This work is also hard to follow, the contribution and the method should be highlighted and stated.
>
> While our paper already includes a dedicated paragraph on the contributions in the introduction, we agree that this paragraph would benefit from more explicit emphasis on the relevant aspects. Therefore, we included a bullet-point list to clearly state the contribution of the method and hope this addresses the reviewer's concerns.
>
> > Please abatement the contribution of this work. This work is not the first to use DT to represent RL policy, i.e., the work "MIXRTs: Toward interpretable multi-agent reinforcement learning via mixing recurrent soft decision trees", arXiv:2305.10091, 2022. They utilized SDT in multi-agent RL complex tasks two years ago.
>
> **TL;DR: SYMPOL does not learn differentiable or soft DTs, but instead maintains univariate, hard decisions during training and inference. A clarifying paragraph has been added to the revised paper.**
>
> Thank you for highlighting additional related work on decision trees in reinforcement learning. However, we emphasize a fundamental distinction between our approach and the referenced work, "MIXRTs: Toward interpretable multi-agent reinforcement learning via mixing recurrent soft decision trees." Unlike soft decision trees (SDTs), which approximate decisions with probabilistic nodes, our method uses hard, axis-aligned DTs that enforce discrete decisions, leading to more interpretable policies. We will add the suggested work to our refined related work section. Additionally, we included a dedicated paragraph to distinguish our contribution from soft and differentiable decision trees (L131; L138-L143).
>
> We also agree with the reviewer that our initial claim may have been misleading. We do not claim that our work is the first to use DTs to represent policies in reinforcement learning. As the reviewer notes, this approach has been extensively explored in prior work. However, existing work typically focuses on SDTs for end-to-end learning. Instead, we utilize hard, axis-aligned DTs while still preserving gradient-based optimization. To clarify, we have amended the claim in the abstract (L20-L23) and expanded the related work section with a more detailed discussion and comparison with existing methods (L138-L143; L247-L252).

---

> ### Author Response · Authors · 2024-11-20
> **Rebuttal (Part 2 / 2)**
>
> > There are some typos in References.
>
> Thank you for highlighting this. We have improved this in the revised version of the paper.
>
> > line 414- 415: "In RL, especially for smaller policies, the runtime is mainly determined by the actor-environment interaction..." - This sentence is hard to follow. Can the authors unpack this sentence?
>
> We have unpacked and clarified the sentence as follows in the revised version of the paper (L428-L431):
> *"In RL, the actor-environment interaction frequently constitutes a significant portion of the total runtime. For smaller policies, in particular, the runtime is mainly determined by the time required to execute actions within the environment to obtain the next observation, while the time required to execute the policy itself having a comparatively minimal impact on runtime."*
>
> > Did this work simply incorporate GradTree into the PPO framework?
>
> **TL;DR: Our approach extends beyond simply incorporating GradTree into PPO by developing essential adjustments that enable a stable, effective, and interpretable model and facilitate seamless integration into existing RL frameworks.**
>
> Thank you for your constructive feedback. While our approach builds on the concept of gradient-based, axis-aligned, hard decision trees (GradTree), we extend it considerably to address the unique challenges of RL. Directly applying GradTree within RL frameworks results in unstable and ineffective outcomes, as the inherent instability in training tree-based models poses distinct difficulties in this setting.
>
> To achieve stability and effectiveness in RL, we developed several essential adjustments. As demonstrated in our ablation study (Figure 5), these modifications are crucial for SYMPOL to retain interpretability while achieving robust performance under end-to-end gradient-based optimization. Without these adjustments, both stability and effectiveness in RL applications would be significantly compromised.
>
> However, with these adaptations in place, SYMPOL is framework-agnostic, allowing for straightforward integration into diverse RL algorithms beyond PPO. As illustrated in Section A.1 of the appendix, SYMPOL integrates seamlessly with A2C, demonstrating superior performance compared to alternative methods. This adaptability highlights the broad applicability and robustness of our approach across RL frameworks. In the revised version of the paper, we discussed the contributions and advantages more thoroughly (L70-L85; L180-L184).
>
>
> `[1]` Wei, Hua, et al. "Intellilight: A reinforcement learning approach for intelligent traffic light control." Proceedings of the 24th ACM SIGKDD international conference on knowledge discovery & data mining. 2018.
>
> `[2]` Yau, Kok-Lim Alvin, et al. "A survey on reinforcement learning models and algorithms for traffic signal control." ACM Computing Surveys (CSUR) 50.3 (2017): 1-38.
>
> `[3]` Yu, Chao, et al. "Reinforcement learning in healthcare: A survey." ACM Computing Surveys (CSUR) 55.1 (2021): 1-36.
>
> `[4]` Nambiar, Mila, et al. "Deep offline reinforcement learning for real-world treatment optimization applications." Proceedings of the 29th ACM SIGKDD Conference on Knowledge Discovery and Data Mining. 2023.
>
> `[5]` Yu, Zheng, et al. "Deep reinforcement learning for cost-effective medical diagnosis." arXiv preprint arXiv:2302.10261 (2023).
>
> `[6]` Fatemi, Mehdi, et al. "Medical dead-ends and learning to identify high-risk states and treatments." Advances in Neural Information Processing Systems 34 (2021): 4856-4870.

---

### Author Response · Authors · 2024-11-20
**Rebuttal Summary**

We greatly appreciate the reviewers' constructive feedback and acknowledgment of our method's strengths and novelty. In particular, the reviewers `[DZ3Q, sgid, 6L9X, 7Ps3, 6tAh]` highlighted the motivation and relevance of our approach, noting its *“huge potential impact for interpretable RL through online learning of trees.”* `[6L9X]`. Furthermore, the reviewers pointed out our method's advantage over alternative approaches that do not directly optimize the policy `[6L9X, 6tAh]`, as well as the clarity of our presentation `[6L9X, 7Ps3, 6tAh]` and the value of the insightful case study `[7Ps3, 6tAh]`.

We summarize below the main concerns raised by the reviewers and explain how we have addressed each one. We have provided detailed responses to all questions directly to each reviewer. Additionally, we have uploaded a revised version of the manuscript, incorporating adjustments based on the reviewers' feedback. We have also included a version of the revised manuscript highlighting all changes, provided in the supplementary material for transparency.

* **Distinguishing SYMPOL from Differentiable Decision Tree Methods** `[DZ3Q, sgid, 7Ps3]`: We revised the related work section to more clearly distinguish SYMPOL from methods that employ differentiable decision trees in a separate paragraph, addressing the reviewers’ feedback regarding this point (L138-L143; L247-L252). In related work, differentiable decision trees, as for example used by Silva et al. (2020), are soft decision trees (SDTs), achieving differentiability by relaxing discrete decisions for feature and path selection at internal nodes. This approach is fundamentally different from SYMPOL, which does **not** use differentiable decision trees. Instead, SYMPOL leverages GradTree to optimize standard, non-differentiable decision trees through gradient descent. As a result, SYMPOL retains univariate, hard decisions, which are more interpretable than the oblique splits involving multiple features in SDTs and differentiable decision trees. Furthermore, SYMPOL does not require any post-processing to obtain interpretable, standard decision trees, which would typically result in a substantial information loss, as demonstrated in Section 5.2 of our paper.

* **Comparison with Related Work** `[6L9X, 7Ps3]`: In the revised version, we clarified the relationship between SA-DT and VIPER  (L343-L346; Section A.3), which was not explicitly outlined in the initial submission. In general, SA-DT can be considered as a version of DAGGER. Therefore, SA-DT and VIPER are conceptually similar, but VIPER (Q-DAGGER) includes advanced sampling and weighting strategies for training samples — techniques not incorporated in SA-DT as used, for example, by Silva et al. (2020). To address this, we implemented VIPER in our evaluation, enabling a direct comparison. The evaluation results remain consistent with those in the initial submission. SYMPOL achieves superior performance in most cases. This is also in-line with the results reported by `[4]`, which demonstrate that the sampling in VIPER does not result in superior performance compared to DAGGER/SA-DT when learning small, interpretable decision trees which is the focus of our paper.

* **Recentering of contributions** `[7Ps3]`: We clarified our contributions which are now centered around the problem we are solving: Mitigating the information loss inherent in existing approaches through a direct optimization of an interpretable decision tree policy with SYMPOL.

* **SYMPOL with (affine) linear combinations of features** `[7Ps3]`: We extended SYMPOL to include (affine) linear features and observed a significant performance increase in environments such as LunarLander, where (affine) linear feature combinations are relevant (reward improvement from -57 to 105). These results suggest that SYMPOL with (affine) linear features is capable of addressing more complex environments. We will add complete experiments with (affine) linear features to the appendix as soon as our experiments including HPO have finished. In addition, we are currently evaluating the performance of SYMPOL with affine linear feature on (Object-Centric) Atari environments, where preliminary results appear promising.

---

### Meta-Review · Area_Chair_jtfM · 2024-12-19

**Metareview:**

This paper presents SYMPOL, integrating tree-based models with policy gradient algorithms for interpretable reinforcement learning policies. The reviewers praised the paper's clear writing and thorough empirical evaluation. Initial concerns about technical novelty and similarity to previous work were addressed through detailed clarifications of SYMPOL's unique integration of GradTree and stability-enhancing modifications. The authors made significant improvements during review, particularly in extending SYMPOL to handle affine linear features and demonstrating its unique approach to maintaining hard decision boundaries versus soft decision trees.
Moving forward, I suggest the authors consider expanding evaluations to more complex environments and providing clearer differentiation from recent work on differentiable decision trees. Given the paper's valuable contribution to interpretable RL and the authors' strong engagement with reviewer feedback through substantive improvements, I recommend acceptance.

**Additional Comments On Reviewer Discussion:**

The reviewers praised the paper's clear writing and thorough empirical evaluation. Initial concerns about technical novelty and similarity to previous work were addressed through detailed clarifications of SYMPOL's unique integration of GradTree and stability-enhancing modifications. The authors made significant improvements during review, particularly in extending SYMPOL to handle affine linear features and demonstrating its unique approach to maintaining hard decision boundaries versus soft decision trees.

---

### Decision · Program_Chairs · 2025-01-22

Accept (Spotlight)